# Lipoprotein lipase regulates hematopoietic stem progenitor cell maintenance through DHA supply

Chao Liu[1], Tianxu Han[2], David L. Stachura[3], Huawei Wang [4], Boris L. Vaisman[5], Jungsu Kim[1], Richard L. Klemke[4], Alan T. Remaley[5], Tariq M. Rana [2], David Traver [3] & Yury I. Miller[1]

Lipoprotein lipase (LPL) mediates hydrolysis of triglycerides (TGs) to supply free fatty acids (FFAs) to tissues. Here, we show that LPL activity is also required for hematopoietic stem progenitor cell (HSPC) maintenance. Knockout of Lpl or its obligatory cofactor Apoc2 results in significantly reduced HSPC expansion during definitive hematopoiesis in zebrafish. A human APOC2 mimetic peptide or the human very low-density lipoprotein, which carries APOC2, rescues the phenotype in apoc2 but not in lpl mutant zebrafish. Creating parabiotic apoc2 and lpl mutant zebrafish rescues the hematopoietic defect in both. Docosahexaenoic acid (DHA) is identified as an important factor in HSPC expansion. FFA-DHA, but not TG-DHA, rescues the HSPC defects in apoc2 and lpl mutant zebrafish. Reduced blood cell counts are also observed in Apoc2 mutant mice at the time of weaning. These results indicate that LPL-mediated release of the essential fatty acid DHA regulates HSPC expansion and definitive hematopoiesis.

[1] Department of Medicine, University of California, San Diego, 9500 Gilman Drive, La Jolla, CA 92093, USA. [2] Department of Pediatrics, University of California, San Diego, 9500 Gilman Drive, La Jolla, CA 92093, USA. [3] Department of Cellular and Molecular Medicine, University of California, San Diego, 9500 Gilman Drive, La Jolla, CA 92093, USA. [4] Department of Pathology, University of California, San Diego, 9500 Gilman Drive, La Jolla, CA 92093, USA. [5] Lipoprotein Metabolism Section, Cardio-Pulmonary Branch, National Heart, Lung, and Blood Institute, 31 Center St, Bethesda, MD 20892, USA. Correspondence and requests for materials should be addressed to Y.I.M. (email: yumiller@ucsd.edu)

Lipoprotein lipase (LPL) is a major lipase in the vasculature responsible for hydrolysis of triglycerides (TGs) carried by TG-rich lipoproteins and supplying free fatty acids (FFAs) to tissues[1]. Apolipoprotein C-II (APOC2) is an obligatory cofactor required for LPL activity[2]. Human patients with APOC2 or LPL deficiency, or deficiency in glycosylphosphatidylinositol-anchored high-density lipoprotein-binding protein 1 (GPIHBP1), the LPL vascular anchor, develop severe hypertriglyceridemia and chylomicronemia[1, 3]. Recent data indicating that plasma TG levels predict cardiovascular risk[4, 5] have revived scientific community's interest in regulation of LPL activity. Study of LPL activity in mice was initially impeded by post-natal lethality of systemic Lpl knockout[6, 7]. Tissue-specific LPL deficiency in adipose tissue resulted in decreased FFA uptake but increased endogenous synthesis of non-essential FFAs[8]. Heart-specific Lpl knockout mice showed cardiac dysfunction despite a compensatory increase in glucose utilization[9]. Similar cardiac phenotypes were observed in human patients with LPL deficiency[10]. Tissue-specific Lpl overexpression studies suggested that LPL is a key enzyme responsible for tissue-specific insulin sensitivity and lipid metabolism[11, 12]. These studies implicate LPL-mediated TG hydrolysis and release of FFAs as a key regulator of many physiologic processes in specific tissue contexts. We recently reported the development of systemic Apoc2 mutant mice, characterized by moderate-to-severe hypertriglyceridemia[13], which will be used in future studies to investigate related phenotypes.

Zebrafish models have emerged as a new powerful tool to study lipid metabolism[14]. A hyperlipidemia response to feeding regimens, cholesteryl ester transfer protein (CETP) expression[15], pliancy to genetic modifications, and the optical transparency of larval zebrafish significantly facilitate these studies. We have recently reported a chylomicronemia and hypertriglyceridemia phenotype in apoc2 knockout zebrafish[16]. In the present study, we have developed an lpl knockout zebrafish, which have a similar hypertriglyceridemia phenotype. Remarkably, both apoc2 and lpl mutant zebrafish display profound anemia and defects in hematopoietic stem progenitor cell (HSPC) maintenance and differentiation. Parabiosis of apoc2 and lpl mutants rescues the defective HSPC expansion in both mutants, indicating the importance of circulating FFAs. Docosahexaenoic acid (DHA) is selectively reduced in apoc2 zebrafish mutants. Injections of exogenous DHA in an FFA form, but not the DHA esterified into a TG, rescues the HSPC defects in apoc2 and lpl mutants. In addition, we report anemia in young Apoc2 mutant mice. These findings may have important therapeutic implications for using DHA as a dietary supplement to treat anemia and/or expand HSCs ex vivo.

## Results

**Loss of apoc2 function in zebrafish results in anemia.** Red blood cells of adult apoc2 mutant zebrafish were characterized by hypochromia and decreased hemoglobin staining (Fig. 1a), and the total blood cell count in apoc2 mutants was significantly lower than in WT zebrafish (Fig. 1b). Decreased blood cell numbers, increased numbers of immature erythrocytes and weak hemoglobin staining were also observed in 6.3 days post-fertilization (dpf) zebrafish larvae, but not in 52 h post-fertilization (hpf) embryos (Fig. 1c–e and Supplementary Movies 1 and 2).

The hematopoiesis phenotype in apoc2 knockout zebrafish could be due to hyperlipidemia, i.e., high levels of non-hydrolyzed TG, and/or due to diminished FFA supply. To test whether hematopoietic defects are a direct result of hyperlipidemia, we treated apoc2 mutants with lomitapide, an inhibitor of microsomal triglyceride transfer protein, which reduces very low-density lipoprotein (VLDL) formation and is used as a lipid-

lowering drug for treatment of familial hypercholesterolemia patients. Lomitapide has been reported to reduce hyperlipidemia in zebrafish as well[17]. Treatment with lomitapide starting from 2 dpf reduced hyperlipidemia but did not rescue anemia in apoc2 mutant zebrafish (Fig. 1f, g). Similarly, feeding apoc2 mutants a low-fat diet reduced hyperlipidemia but did not have a significant effect on anemia (Supplementary Fig. 1 and Fig. 1h).

**lpl mutants show defective hematopoiesis.** Zebrafish apoc2 is expressed in the yolk and intestine, but lpl is expressed in the head and in the caudal hematopoietic tissue (CHT) at 2 dpf, the latter is a hematopoietic organ during embryonic development (Supplementary Fig. 2 and Fig. 2a). To further evaluate the role of LPL activity in hematopoiesis, we mutated the zebrafish lpl gene using CRISPR-Cas9 and obtained a line with a 2 nt deletion in exon 4 of the lpl gene, resulting in-frame-shift and a pre-stop codon. This mutation may result in a truncated mRNA or in-frame exon skipping and alternative splicing[18]. To address the possibility of an alternative splicing, we cloned cDNA from the lpl mutant and found no alternative transcripts. Furthermore, we confirmed the presence of only one transcript carrying the 2 nt deletion that encodes a truncated, loss-of-function Lpl protein lacking its heparin-binding domain (Fig. 2b and Supplementary Fig. 3A–C). Interestingly, there was a compensatory increase in apoc2 expression in the lpl mutants, and a compensatory increase in lpl expression in the apoc2 mutants (Fig. 2c). However, because APOC2 does not have a catalytic activity and LPL is not functional without APOC2, both mutants developed hypertriglyceridemia (Fig. 2d and Supplementary Fig. 3D–E). Importantly, we found a similar anemia and hypochromia phenotype in adult and 6.3 dpf larval lpl mutants as those found in apoc2 mutants (Fig. 2e–i).

**HSPC expansion is disrupted in apoc2 and lpl mutants.** Similar to mammals, zebrafish have two major waves of hematopoiesis, primitive and definitive[19–21]. The transitive primitive hematopoiesis begins at 11 hpf and produces erythroid and myeloid cells from mesoderm-derived hemangioblasts that persist for the first several days post fertilization. The definitive hematopoiesis is attained via HSPCs, which are specified from hemogenic endothelium comprising the ventral aspect of the dorsal aorta (VDA) at 20 hpf and then migrate to the CHT, where HSPCs undergo expansion and differentiate into mature blood cells at 2–4 dpf.

Interestingly, apoc2 and lpl mutants did not have any detectable defects in total blood cell count or hemoglobin content at 52 hpf, but these measurements were decreased at 6.3 dpf (Figs. 1c–e and 2i). Because erythrocytes are mostly derived from primitive hematopoiesis at 52 hpf and largely from definitive hematopoiesis at 6.3 dpf, we hypothesized that loss of apoc2 may affect definitive hematopoiesis and thus cause anemia at later larval stages and in adults. To test this hypothesis, we performed whole mount in situ hybridization with probes for specific hematopoietic markers at different stages.

At 20 hpf, wild-type and apoc2 mutant zebrafish had similar expression of gata1 and beta-globin (erythropoiesis), and pu.1 (myelopoiesis) (Supplementary Fig. 4A), suggesting no detectable defects in primitive hematopoiesis. Furthermore, wild-type and apoc2 mutants showed similar expression of HSPC markers runx1 in the VDA at 26 hpf and cmyb in the VDA and CHT at 52 hpf (Supplementary Fig. 4B), implicating that HSPC specification and migration were not affected. However, at 80 hpf, expression of the HSPC markers runx1/cmyb and the blood lineage markers beta-globin (erythropoiesis) and rag1 (lymphopoiesis) was significantly decreased in apoc2 mutants compared to wild type (Supplementary Fig. 4C), suggesting that HSPC expansion and

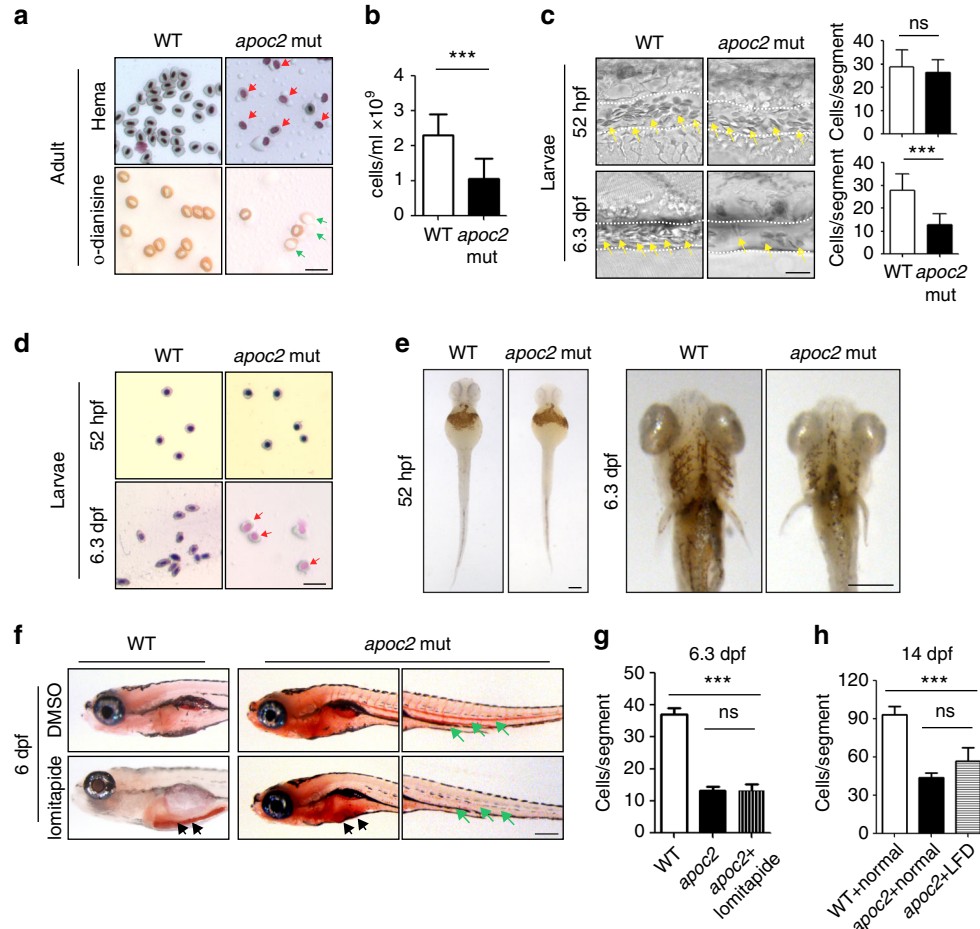

**Fig. 1** Anemia in *apoc2* mutant zebrafish. **a** Wright–Giemsa (Hema) and o-dianisine staining of peripheral blood cells from adult (18–20-month-old) male wild-type (WT) and *apoc2* mutant zebrafish. **b** Quantitative results of peripheral blood cell count (*n* = 11 for WT and *n* = 9 for *apoc2* mutant groups). **c** Representative bright field images and quantitative results of blood cell (yellow arrows) count in the caudal vein (outlined with white dashed lines) of WT and *apoc2* mutants at 52 hpf (*n* = 8 in WT and *n* = 9 in *apoc2* mutant groups) and 6.3 dpf (*n* = 11 for WT and *n* = 13 for *apoc2* mutant groups). See also Supplementary Movies 1 and 2. **d** Wright–Giemsa staining of blood smears from WT and *apoc2* mutants at 52 hpf and 6.3 dpf. Immature erythrocytes containing larger and less condensed nuclei are indicated with red arrows in **a** and **d**. Immature erythrocytes with weaker hemoglobin staining are indicated with green arrows in **a**. **e** o-Dianisine staining of 52 hpf and 6.3 dpf WT and *apoc2* mutant embryos. **f** Effect of lomitapide: WT and *apoc2* mutants were treated with 5 μM lomitapide starting from 2 dpf until embryos were fixed at 6 dpf for ORO staining. Black arrows point to intestinal lipid accumulation and green arrows to circulating lipids. **g** Blood cell counts in the caudal vein of WT, *apoc2* mutants and *apoc2* mutants treated with lomitapide at 6.3 dpf (*n* = 5 in each group). **h** Blood cell counts in the caudal vein of 14 dpf WT, *apoc2* mutants fed with normal diet and *apoc2* mutants fed with low-fat diet (LFD) starting at 5 dpf (*n* = 4 in WT and *apoc2* mut groups each; *n* = 5 in *apoc2* + LFD group). Scale bars, 20 μm in **a**, **c**, and **d**, 100 μm in **e**, and 200 μm in **f**. Mean ± SEM; ***P < 0.001 (Student's *t* test)

differentiation were affected. The HSPC expansion defect was not due to a moderately delayed angiogenesis in the *apoc2* mutant[16], as the defects persisted in angiogenesis-synchronized *apoc2* mutant embryos (Supplementary Fig. 5). As expected from results shown in Fig. 1g, lomitapide did not rescue the defect in *runx1/cmyb* and *beta-globin* expression in *apoc2* mutants (Supplementary Fig. 6). There was no increase in apoptosis in the CHT region of *apoc2* mutants (Supplementary Fig. 7). Comparing *apoc2* mutants with *lpl* mutants, we confirmed that in both animal models, the hematopoietic defect commenced at the HSPC expansion (80 hpf) but not the specification or migration (26–50 hpf) stage of definitive hematopoiesis (Fig. 3a, b).

To visualize HSPCs in real-time, we used *cd41*:EGFP transgenic zebrafish in which HSPCs display weak EGFP fluorescence and become EGFP bright upon differentiation into thrombocyte progenitors in the CHT[22, 23]. In agreement with the in situ hybridization results, at 54 hpf, there were similar number of EGFP[low] cells in the CHT region of wild-type and *apoc2*

mutants. When the same clutch of zebrafish was re-examined at 80 hpf, the numbers of HSPCs (EGFP[low]) were significantly decreased in the CHT region of *apoc2* mutants when compared to wild type (Fig. 3c, d). Consistent with the CHT data, numbers of EGFP-positive HSPCs that migrated from the CHT to the thymus, where they differentiate into lymphoid cells, decreased dramatically from 54 hpf to 4 dpf (Fig. 3e, f).

**LPL-mediated triglyceride hydrolysis regulates hematopoiesis.**
If the hematopoietic defect in *apoc2* mutants manifests from 50 hpf to 80 hpf, then restoration of *apoc2* function starting at 2 dpf should rescue the hematopoiesis phenotype. Indeed, injecting the human APOC2 mimetic peptide CII-a, but not its inactive analog CII-i, rescued the hyperlipidemia and anemia phenotype at 6.3 dpf (Fig. 4a–d) and, importantly, restored both *cmyb/runx1* and *beta-globin* expression at 3.3 dpf (Fig. 4e) in *apoc2* mutants.

We next investigated whether TG hydrolysis, irrespective of hypertriglyceridemia, could affect normal hematopoiesis.

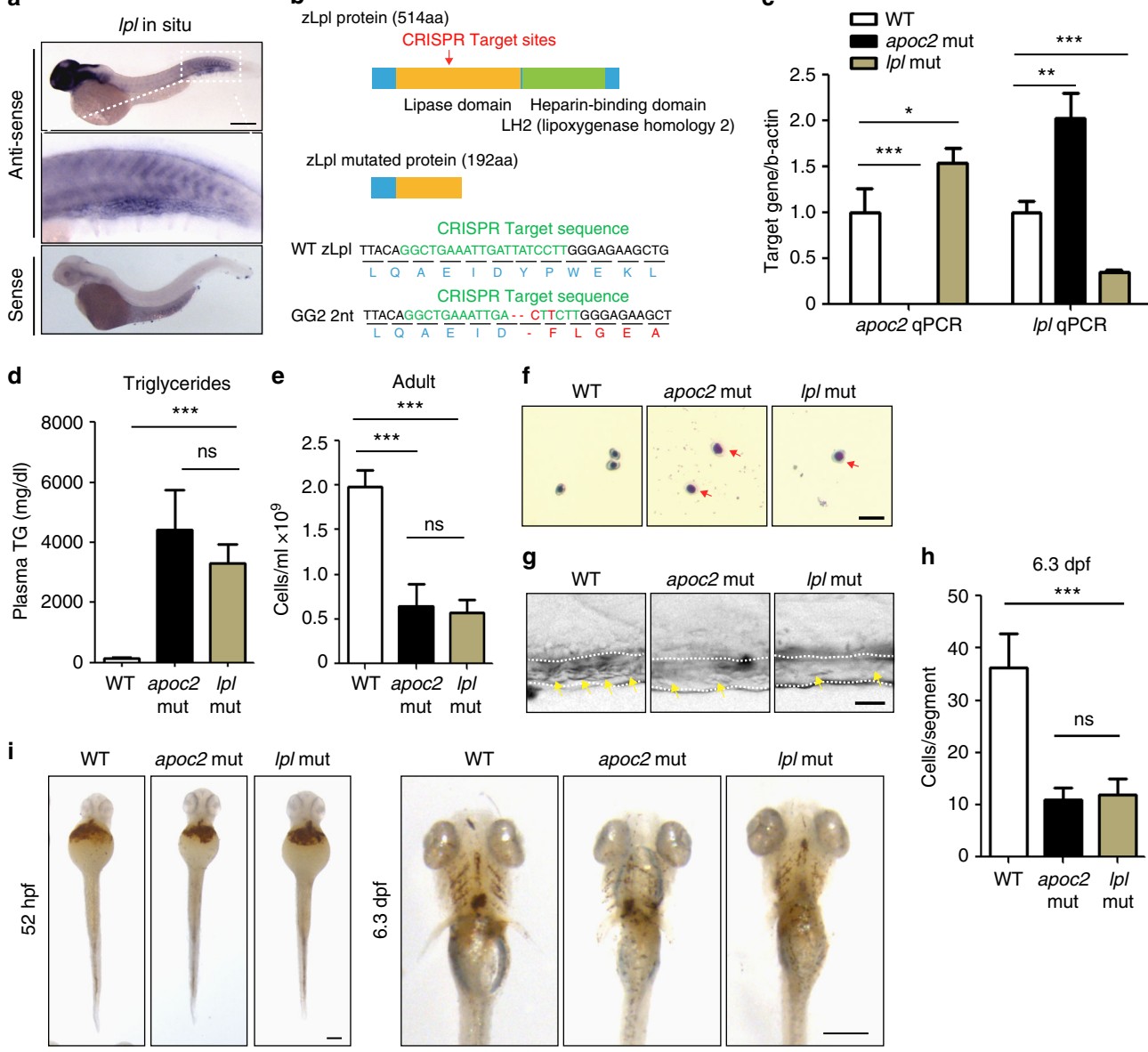

**Fig. 2** Hematopoietic defects in *lpl* mutants. **a** In situ hybridization with *lpl* antisense and sence probes in WT embryos at 2 dpf. **b** Diagram of *lpl* CRISPR target sites and the predicted truncated protein caused by the 2 nt deletion, which results in a codon shift and premature translation termination. **c** qPCR results of *lpl* and *apoc2* mRNA expression in WT, *apoc2* and *lpl* mutants at 5 dpf (*n* = 3 in each group). **d** Plasma TG levels in adult (9–15-month-old) male WT, *apoc2* and *lpl* mutants (*n* = 5 in each group). **e** Peripheral blood cell count in adult (15-month-old) male WT, *apoc2* and *lpl* mutants (*n* = 5 in each group). **f** Wright–Giemsa staining of blood smears from 6.3 dpf WT, *apoc2* and *lpl* mutants. **g**, **h** Representative images and quantitative results of blood cell (yellow arrows) count in the caudal vein (outlined with white dashed lines) of WT, *apoc2* and *lpl* mutants at 6.3 dpf (*n* = 5 in WT and *apoc2* mut groups each; *n* = 6 in *lpl* mut group). **i** o-Dianisine staining of 52 hpf and 6.3 dpf WT, *apoc2* and *lpl* mutant embryos. Scale bars, 200 μm in **a**; 20 μm in **f**, **g**, and 100 μm in **i**. Quantitative results are mean ± SEM; *$P < 0.05$ and ***$P < 0.001$ (Student's *t* test)

Injection of human VLDL into WT embryos at 2 dpf enhanced expression of HSPC and erythrocyte markers at 80 hpf, suggesting that oversupply of the TG substrate for Lpl-mediated hydrolysis promotes, rather than inhibits, HSPC expansion and erythroid differentiation in the CHT region at this specific stage (Fig. 4f). Importantly, VLDL (which carries APOC2), but not LDL (which does not contain APOC2), rescued the defective hematopoiesis in *apoc2* mutants when injected at 2 dpf (Fig. 4f). However, VLDL injection did not rescue the hematopoiesis defects in *lpl* mutants (Fig. 4f) because these zebrafish did not have functional Lpl to be activated by VLDL-delivered APOC2.

Together, these data suggest that hyperlipidemia per se does not cause a hematopoietic defect in *apoc2* mutants and that

ineffective plasma TG hydrolysis in *apoc2* and *lpl* mutants is the cause of defective HSPC expansion and differentiation in the CHT region.

HSPCs anchor to mesenchymal stromal cells after arrival in the CHT region and this interaction is crucial for HSPCs maintenance and differentiation[24]. To determine which cell types express *lpl*, we sorted out HSPCs and stromal cells from the zebrafish expressing EGFP driven by the *gata2* promoter (HSPCs) and mCherry driven by *sdf1α* promoter (stromal cells; Fig. 5a, b). The sorting results were confirmed by testing mRNA expression of *runx1* and *sdf1a* in isolated fractions (Fig. 5c). Because there were 100-fold more stromal cells than HSPCs (Fig. 5b) and because stromal cells, but not HSPCs, highly expressed *lpl*

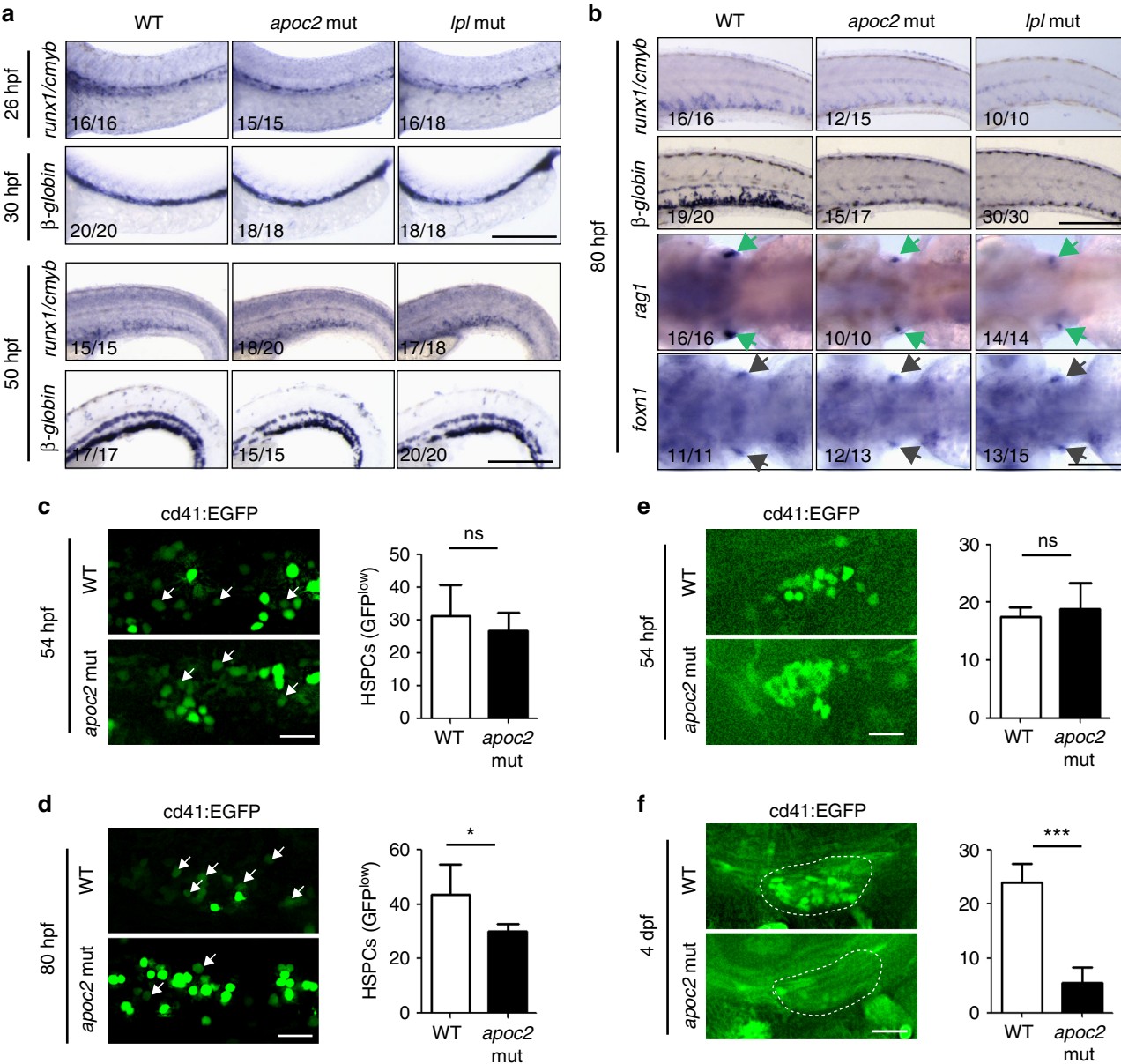

**Fig. 3** Hematopoietic defects in *apoc2* and *lpl* mutants occur during HSPC expansion. **a** In situ hybridization with *cmyb/runx1* and *β-globin* probes in WT, *apoc2* and *lpl* mutants at 26 hpf, 30 hpf and 50 hpf. **b** In situ hybridization with *cmyb/runx1*, *β-globin*, *rag1* (green arrows) and *foxn1* (black arrows) probes in WT, *apoc2* and *lpl* mutants at 80 hpf. *foxn1* is a thymus development marker, used as a control. **c**, **d** Representative images and numbers of GFP^low cells (HSPCs, white arrows) in the CHT region of *cd41*:EGFP transgenic WT and *apoc2* mutants at 54 and 80 hpf ($n = 10$ in WT and $n = 8$ in *apoc2* mutant groups at 54 hpf; $n = 6$ in each group at 80 hpf). **e**, **f** Representative images and numbers of GFP-positive cells in the thymus region at 54 hpf ($n = 9$ in WT and $n = 8$ in *apoc2* mutant groups) and at 4 dpf ($n = 8$ in each group). Scale bars, 200 μm in **a** and **b**; 50 μm in **c**–**f**. Mean ± SEM; *$P < 0.05$ and ***$P < 0.001$ (Student's $t$ test)

(Fig. 5c), we concluded that stromal cells are the major source of *lpl* expression in the CHT.

To characterize in vivo the requirement for both Apoc2 and Lpl in supporting hematopoiesis, we employed a parabiotic zebrafish protocol[25]. After fusion of a *sdf1α*:mCherry or a *ahmc*: EGFP transgenic embryo, in which the EGFP is expressed in the cardiac and skeletal muscle cells at early stages, with a WT embryo, we found no detectable tissue expression of mCherry or EGFP in the WT embryo, indicating no stromal or muscle cell exchange (Fig. 5d, e). Fusion of WT embryos with either *apoc2* or *lpl* mutants rescued the hyperlipidemia phenotype in both (Fig. 5f, g), suggesting shared blood circulation in parabiotic zebrafish. Interestingly, fusing *apoc2* mutants with *lpl* mutants

rescued *cmyb/runx1* expression in the CHT region of both (Fig. 5h). These data suggest that under parabiotic settings, Apoc2 from the *lpl* mutant and Lpl from the *apoc2* mutant could reconstitute the Apoc2/Lpl pathway. We thus propose that the Apoc2/Lpl-mediated release of FFAs into shared circulation rescues hematopoiesis in both mutants, even though the Lpl deficiency persists in the CHT niche in *lpl* mutants (Fig. 5i).

**Free fatty acid DHA regulates hematopoiesis in zebrafish.** Although our data suggest that LPL-mediated hydrolysis of TGs regulates HSPC proliferation in zebrafish, surprisingly, we did not

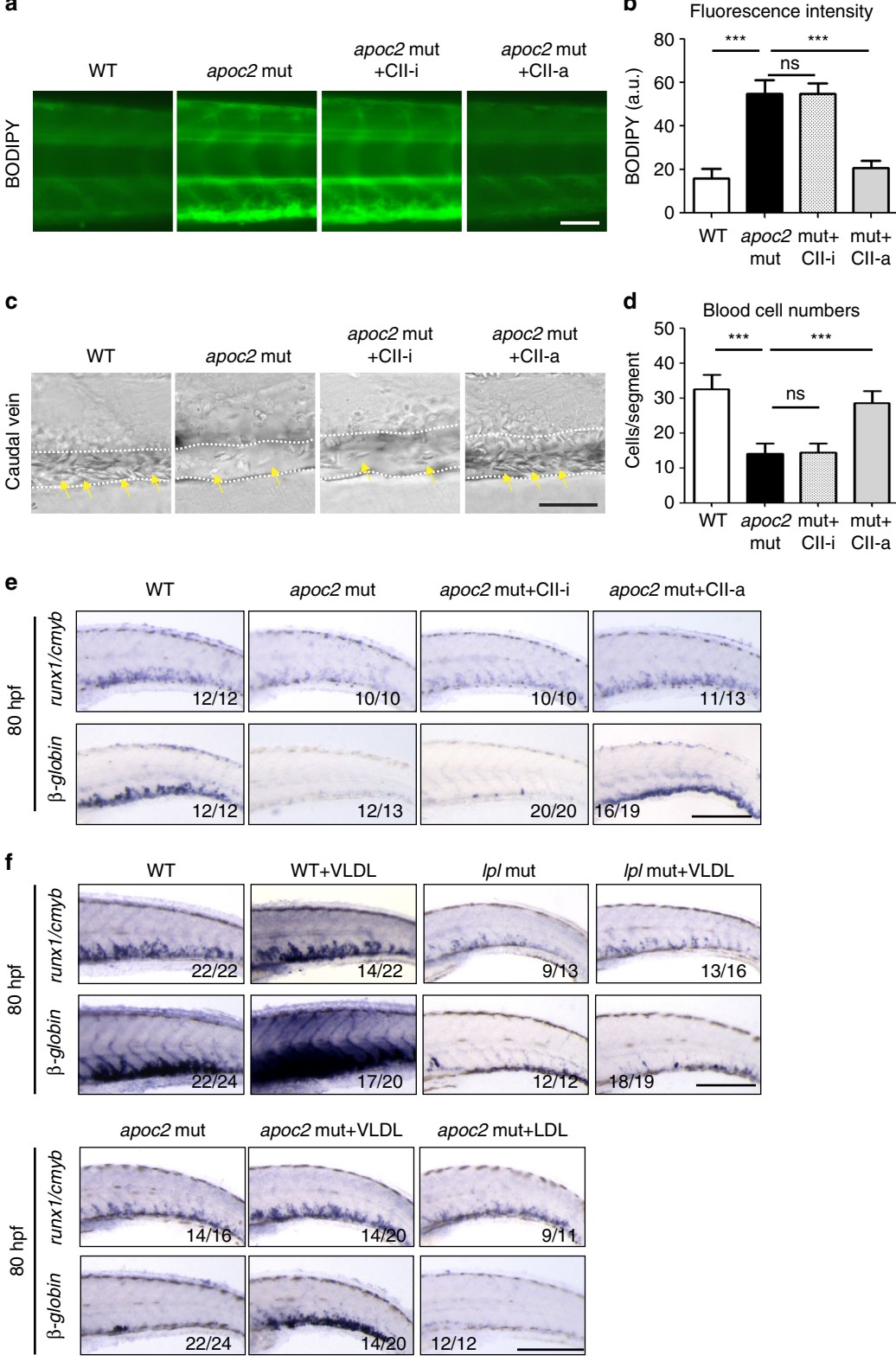

**Fig. 4** APOC2 mimetic peptide and VLDL rescue anemia in *apoc2* mutants. **a**, **b** Representative images of BODIPY staining and quantitative results of BODIPY fluorescence intensity in WT, *apoc2* mutants and the *apoc2* mutants injected with APOC2 mimetic peptides (CII-a, active; CII-i, inactive) at 6.3 dpf (*n* = 5 in each group). **c**, **d** Representative images of blood cells in the caudal vein region and quantitative results of blood cell count in WT, *apoc2* mutants and the *apoc2* mutants injected with APOC2 mimetic peptides at 6.3 dpf (*n* = 6 in WT and mut + CII-i goups each; *n* = 8 in *apoc2* mut group; *n* = 7 in mut + CII-a group). **e** In situ hybridization with *cmyb/runx1* and *β-globin* probes in WT, *apoc2* mutants and the *apoc2* mutants injected with APOC2 mimetic peptides at 80 hpf. Embryos were injected with peptides at 2 dpf. **f** In situ hybridization with *cmyb/runx1* and *β-globin* probes in WT, *apoc2* and *lpl* mutants, including those injected with VLDL or LDL at 2 dpf. Embryos were analyzed at 80 hpf. Scale bars, 50 μm in **a** and **c**; 200 μm in **e** and **f**. Mean ± SEM; ***P < 0.001 (Student's *t* test)

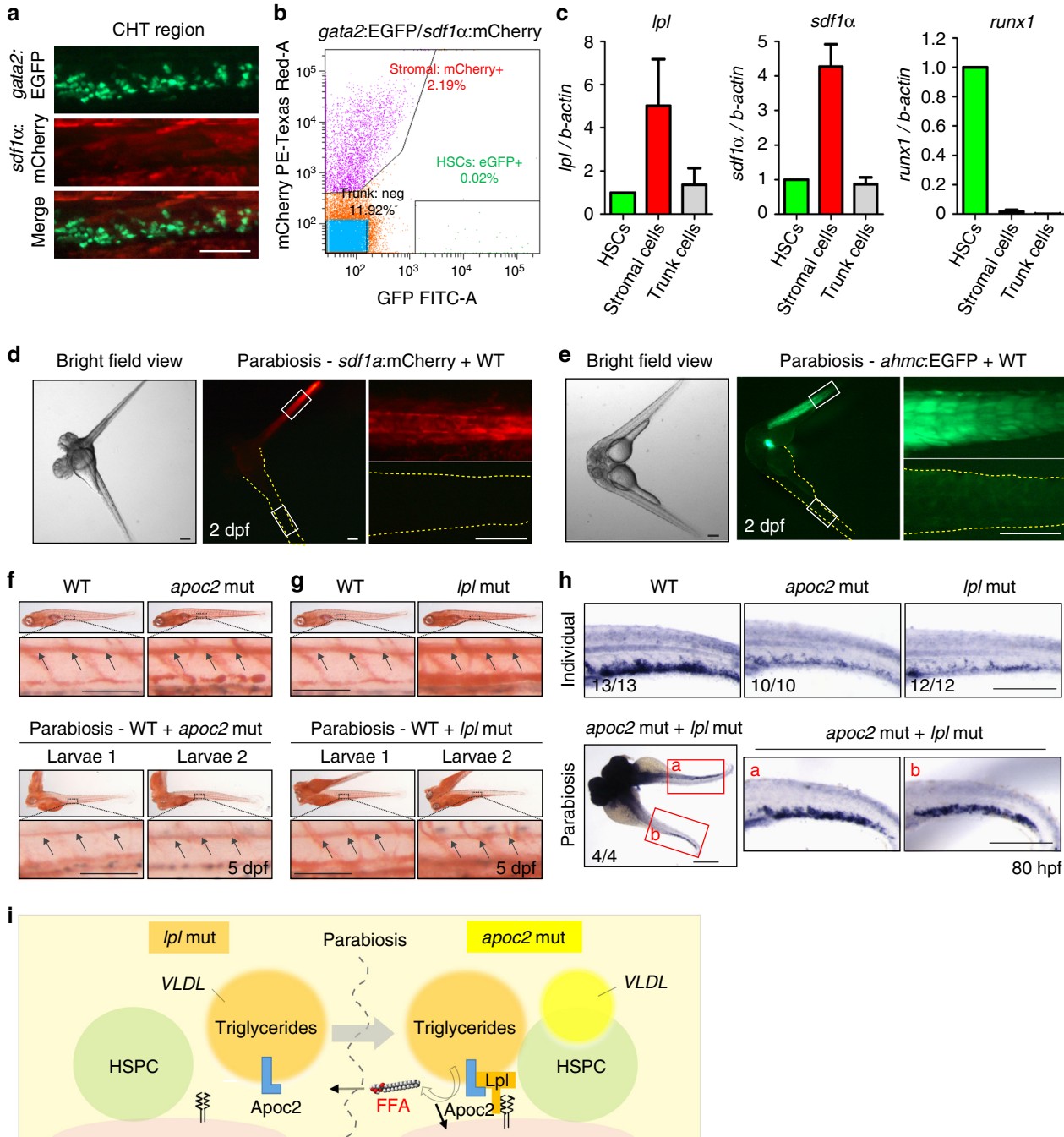

**Fig. 5** Parabiosis of *apoc2* and *lpl* mutants rescues defective hematopoiesis. **a** The CHT region of *gata2*:EGFP, *sdf1a*:mCherry double-positive embryos at 2.5 dpf. **b** Flow cytometry of *gata2*:EGFP and *sdf1a*:mCherry positive cells isolated from the CHT region. **c** RT-qPCR analysis of FACS-sorted *gata2*:EGFP and *sdf1a*:mCherry positive cells, using *lpl*, *runx1* and *sdf1α* primers. Mean ± SD of two independent experiments. **d**, **e** Parabiosis of a *sdf1α*:mCherry or a *ahmc*: EGFP with a WT embryo. Right-hand panels are enlarged segments showed in white quadrangles in left-hand panels. Yellow dashed lines trace WT embryos' boundaries. **f**, **g** Rescue of hyperlipidemia in *apoc2* or *lpl* mutants by parabiosis with WT embryos. Upper panels: two separated embryos (WT and *apoc2* or *lpl*). Lower panels: larva 1 and larva 2 from a parabiosis pair (WT with *apoc2* or WT with *lpl*). Black arrows point to ORO staining in the lumen of blood vessels. **h** In situ hybridization with *runx1/cmyb* probe in WT, individual *apoc2* and *lpl* mutants, and parabiotic *apoc2* and *lpl* mutants at 80 hpf. Scale bars, 50 μm in **f**, **g**, and 200 μm in **a**, **d**, **e** and **h**. **i** Diagram of lipoprotein metabolism in parabiotic *lpl* and *apoc2* mutants. In the *lpl* mutant, no Lpl is expressed. However, VLDL secreted by the *lpl* mutant (orange) delivers Apoc2 through the shared circulation to the *apoc2* mutant, in which Lpl is expressed but its own VLDL (yellow) contains no Apoc2. VLDL from the *lpl* mutant compensates lack of Apoc2 in the *apoc2* mutant and the reconstituted Apoc2/Lpl catalyzes hydrolysis of TG to release FFAs into the shared circulation, which in turn rescue the hematopoiesis defect in both mutants

find a reduction in the total levels of FFAs in total body homogenates of *apoc2* mutants compared to WT (Supplementary Fig. 8, bottom-right graph). This can be explained by a compensatory FFA synthesis in *apoc2* mutants. Indeed, levels of FFAs

that can be made de novo, trended higher or were not changed in *apoc2* mutants. However, levels of DHA (22:6n3), which in zebrafish can only be derived from TG hydrolysis, were significantly lower in 26 hpf embryos (Fig. 6 and Supplementary Fig. 8).

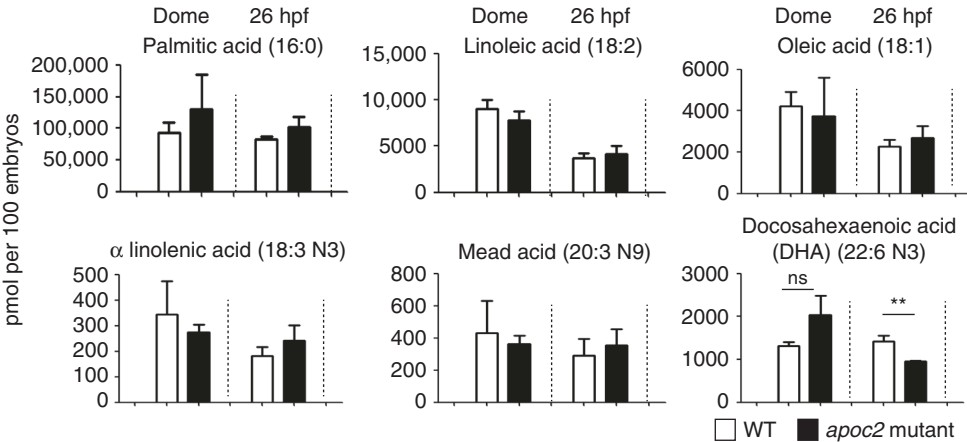

**Fig. 6** Reduced DHA levels in *apoc2* mutants. FFAs in WT and *apoc2* mutants at dome and 26 hpf stages (*n* = 3; each replicate is a pool of 20 embryos); mean ± SEM; **P < 0.01 (Student's *t* test)

To investigate whether DHA plays a specific role in HSPC expansion in the CHT region, we injected DHA, as well as another n3 polyunsaturated fatty acid (PUFA) eicosapentaenoic acid (EPA; 20:5n3) or monounsaturated oleic acid (OA; 18:1n9) (all solubilized with BSA) into *apoc2* mutants at 2 dpf. DHA, but not OA or EPA, rescued red blood cell hypochromia, hemoglobin levels, blood cell counts, and expression of *runx1/cmyb* and *beta-globin* (Supplementary Fig. 9 and Fig. 7a–c). To test whether DHA must be in a free fatty acid form (FFA-DHA) in order to rescue the hematopoiesis phenotype, in a separate experiment, we injected embryos with FFA-DHA or with tridocosahexaenoin, a TG with three esterified DHA acyl chains (TG-DHA; delivered as POPC micelles). FFA-DHA enhanced expression of *runx1/cmyb* and *beta-globin* in WT embryos and rescued hematopoietic phenotypes in *apoc2* and *lpl* mutants (Fig. 8a). Remarkably, injection of TG-DHA enhanced *runx1/cmyb* and *beta-globin* expression only in WT zebrafish but did not have any effect on the phenotypes of *apoc2* or *lpl* mutants (Fig. 8b), which are defective in TG hydrolysis. POPC alone did not have any effect on *runx1/cmyb* and *beta-globin* expression in WT zebrafish (Supplementary Fig. 10). Taken together, these results suggest that DHA is a major functional FFA released from TGs through the APOC2/LPL pathway that plays an important role in HSPC expansion during definitive hematopoiesis in zebrafish (Fig. 8c).

**Anemia in *Apoc2* mutant mice.** In mouse bone marrow, HSPCs reside in a complex niche, including stromal cells[26]. In in vitro experiments, stromal cells are used to support HSPCs maintenance, proliferation, and differentiation. While APOC2 is associated with circulating VLDL, LPL is a cell surface-associated protein highly expressed in tissues dependent on FFA supply for energy (cardiac and skeletal muscle) or storage (adipose)[27–29]. To examine which bone marrow cell types express *Lpl*, we performed RT-qPCR and found that *Lpl* was not expressed in mouse HSPCs (defined for the purposes of this work as Lin⁻cKit⁺ Sca1⁺) or in other Lin⁻ bone marrow cells (Supplementary Fig. 11). However, in agreement with the zebrafish FACS sorting/RT-qPCR data (Fig. 5b, c), stromal cells capable of HSPC support (OP9 cell line) expressed high levels of LPL (Supplementary Fig. 11). Thus, the effect of LPL activity on HSPC maintenance is non-cell autonomous.

While whole-body LPL knockout mice die soon after birth, homozygous *Apoc2* mutant mice, in which three amino acids are deleted, survive and develop hypertriglyceridemia (757.5 ± 281.2 mg/dl)[13]. We found that *Apoc2* mutant mice had much less white

and red blood cells compared to wild-type mice, when assessed at the time of weaning (4 weeks) (Fig. 9a). However, the profound anemia phenotype was alleviated in adult *Apoc2* mutant mice (3–4 months), though mutant mice still had significantly less white blood cells (Fig. 9b). These data suggest that LPL activity may have an important role in mammalian hematopoiesis as well.

**Discussion**

Our work with *apoc2* and *lpl* mutant zebrafish resulted in an observation of profound anemia in these LPL activity deficient zebrafish (Figs. 1 and 2), which was not previously discovered in relevant LPL mouse models. We found that although primitive hematopoiesis and early HSPC specification and migration were not affected in the *apoc2* and *lpl* mutants, there was decreased HSPC expansion in the CHT niche (Fig. 3a, b). With the *cd41: EGFP* transgenic fish line, we found HSPCs, but not differentiated thrombocytes, were decreased during its expansion in the CHT region, which resulted in decreased HSPCs migration to the thymus (Fig. 3c–f). Restoration of Apoc2 function starting at 2 dpf rescued the hematopoietic defects (Fig. 4a–e), confirming the late onset of the requirement for LPL activity in hematopoiesis. The CHT niche is highly vascularized, which facilitates delivery of TG by VLDL. VLDL also carries Apoc2, the activator of Lpl. Thus, injections of human VLDL, supplying both the TG substrate and the APOC2, rescued the HSPCs defect in *apoc2* mutants but not in *lpl* mutants, and increased hematopoietic markers in WT zebrafish (Fig. 4f).

Our in situ hybridization data indicated that *lpl* was highly expressed in the CHT region (Fig. 2a) and FACS sorting/RT-qPCR results suggested stromal cells were the major source of *lpl* expression (Fig. 5a–c). In agreement with the zebrafish results, mouse studies demonstrated that stromal cells, but not HSPCs, expressed *Lpl* (Supplementary Fig. 11), suggesting that HSPCs rely on systemic and/or stromal cell-derived LPL activity. The non-cell autonomous character of LPL activity in regulation of HSPC maintenance complicates mouse studies because of the post-natal lethality of systemic *Lpl* knockout. Although mice with the complete loss of APOC2 are not available, *Apoc2* mutant mice with a three amino acid deletion, which leads to the retention of the signal peptide and the lack of APOC2 on TG-rich lipoproteins, are viable and show moderate hypertriglyceridemia[13], suggesting this mutation is hypomorphic and that *Apoc2* mutant mice can be used as a model of partial LPL deficiency. The finding of anemia in young *Apoc2* mutant mice (Fig. 9a) suggests that our

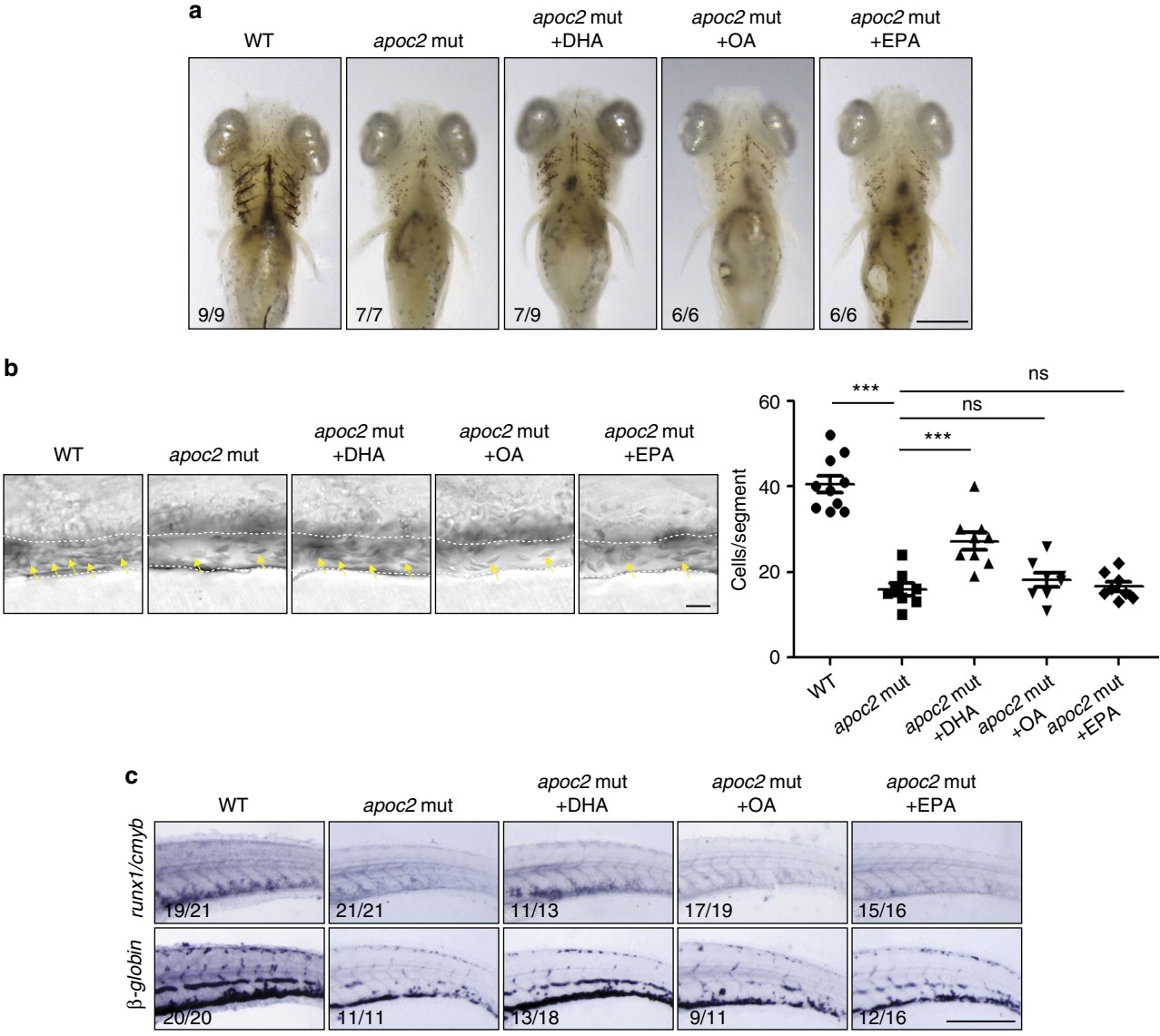

**Fig. 7** DHA rescues hematopoiesis in *apoc2* mutants. **a**–**c** *apoc2* mutant embryos were injected with free fatty acid docosahexaenoic acid (DHA), oleic acid (OA), or eicosapentaenoic acid (EPA) at 48 hpf. **a** o-Dianisine staining of 6.3 dpf larvae. **b** Representative bright field images and quantitative results of blood cell (yellow arrows) count in the caudal vein (outlined with white dashed lines) at 6.3 dpf. Mean ± SEM; $n = 10$ (WT), $n = 8$ (*apoc2* mut, *apoc2* mut + OA, and *apoc2* mut + EPA), and $n = 9$ (*apoc2* mut + DHA). ***$P < 0.001$ (Student's *t* test). **c** In situ hybridization with *cmyb/runx1* and *β-globin* probes. Scale bars, 100 μm in **a**; 50 μm in **b**; and 200 μm in **c**

discovery of the role of LPL activity in zebrafish hematopoiesis is relevant to mammalian biology. There is a report of lower hemoglobin levels and anemia observed in 10 out of 14 infants with LPL deficiency[30], but to the best of our knowledge, there are no reports of anemia in adult human patients with LPL or APOC2 deficiency. This is in agreement with our findings of the anemia phenotype being resolved in adult *Apoc2* mutant mice (Fig. 9b). One reason for anemia resolution in adults could be an increased expression of other lipases, such as hepatic lipase, endothelial lipase, or phospholipase A2, which hydrolyze TG or phospholipid substrates, releasing FFA-DHA into the circulation to support HPSC maintenance. Changes in diet, from fat rich to carbohydrate rich, in mammals, but not in zebrafish may also play a role.

Previous studies have reported a moderate delay in developmental angiogenesis in *apoc2* mutant or knockdown zebrafish embryos, which rapidly recovered in older larvae[16, 31]. It has been

suggested that increased levels of apoB negatively regulate developmental angiogenesis in zebrafish[31]. To evaluate the possibility that the hematopoietic defects observed in *apoc2* mutants may be due to the increased apoB-driven delay in angiogenesis in hyperlipidemic zebrafish, we synchronized angiogenesis in WT and *apoc2* mutant embryos and found that the hematopoietic defects persisted in *apoc2* mutants (Supplementary Fig. 5). Furthermore, inhibition of apoB lipoprotein production by lomitapide did not rescue anemia in *apoc2* mutants (Supplementary Fig. 6) in which LPL function remained defective. These results suggest that the LPL deficiency and not delayed angiogenesis or increased apoB levels is responsible for the hematopoietic defect in *apoc2* mutants. Restoring LPL activity in parabiotic *apoc2* and *lpl* mutants rescued the hematopoietic defect in both (Fig. 5).

We found that the essential fatty acid DHA was specifically downregulated in *apoc2* mutant embryos (Fig. 6 and Supplementary Fig. 8). This is in agreement with a report that DHA

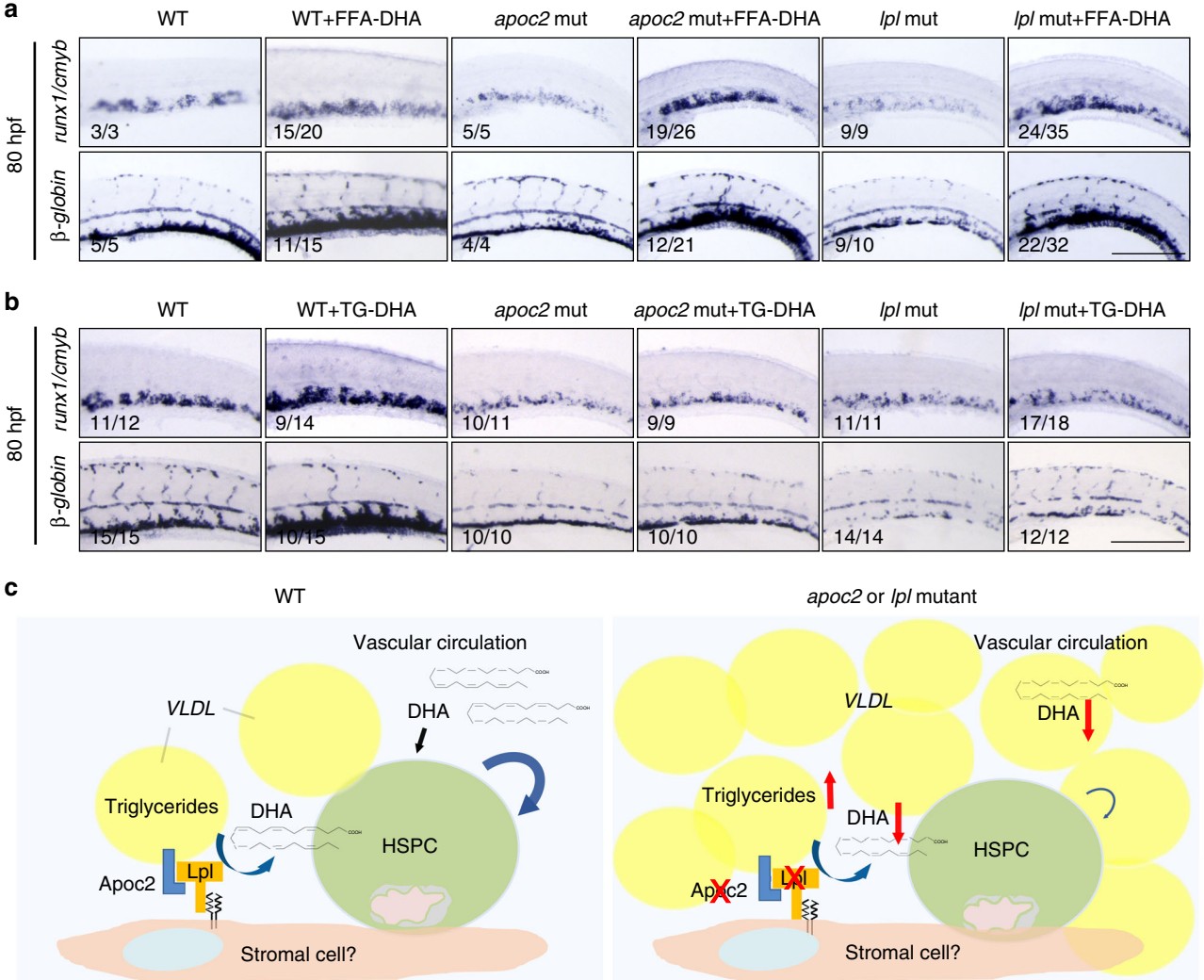

**Fig. 8** FFA-DHA but not TG-DHA rescues hematopoiesis in *apoc2* and *lpl* mutants. **a**, **b** In situ hybridization with *runx1/cmyb* and *β-globin* probes in WT, *apoc2* and *lpl* mutants injected with FFA-DHA (**a**) or TG-DHA (**b**) at 2 dpf; embryos were fixed at 80 hpf for in situ hybridization. Scale bars, 200 μm. **c** Schematic representation of the working hypothesis. VLDL delivers both the TG substrate and Apoc2, an obligatory activator of Lpl to the hematopoietic niche. Lpl, expressed on stromal and/or endothelial cells, catalyzes hydrolysis of TG to produce FFAs. Among FFAs released by the Apoc2/Lpl catalysis, the essential fatty acid DHA supports normal hematopoiesis. *apoc2* and *lpl* mutant zebrafish in which TG hydrolysis is blocked, have a defect in HSPC maintenance and differentiation. Administration of DHA as a free fatty acid, but not DHA esterified into a TG, rescues the hematopoiesis defect in *apoc2* and *lpl* mutant zebrafish

levels were decreased in the hypothalamus of neuron-specific LPL-deficient mice[32]. DHA supplementation increases ex vivo expansion of CD34+ cells derived from umbilical cord or peripheral blood and enhances generation of megakaryocytes[33, 34]. Diets enriched in fish oil have been reported to promote hematopoiesis in mice, the effect that the authors attributed to MMP12-dependent remodeling of the hematopoietic niche[35]. Maternal dietary supplementation of n3 PUFAs, which contain 56% of DHA, has been shown to increase numbers of CD34+ hematopoietic progenitor cells[36]. A clinical trial is under way to test whether diet supplementation with DHA during the second to third trimester of pregnancy could improve the viability of stem cells derived from umbilical cord blood[37].

However, as we now understand from our studies, supply of the essential fatty acid DHA or its precursors with the diet is necessary but not sufficient to promote hematopoiesis. The DHA needs to be present in an FFA form to be active in the hematopoietic niche, but diet-supplied DHA is esterified in

the digestive system to be transported by TG-rich lipoproteins[38]. Thus, LPL activity is necessary to release DHA from the TGs. This is demonstrated in our experiments showing that injections of DHA in the form of FFA, but not in the esterified TG form, rescue the hematopoietic defects in *apoc2* and *lpl* mutants (Fig. 8a, b). Both FFA and TG forms of DHA promote hematopoiesis in the WT zebrafish in which Lpl function is preserved (Fig. 8). The effect seems to be specific to DHA since another n3 PUFA, EPA, or the monounsaturated OA did not rescue hematopoietic defect in *apoc2* mutants (Fig. 7 and Supplementary Fig. 9).

The exact cellular mechanism of FFA-DHA-mediated HSPC maintenance remains to be elucidated. It may include activation of PPARα/δ or upregulation of genes involved in FAO and mitochondrial biogenesis[39–42]. In addition, enzymatic oxidation of DHA might be important for its function since several enzymes capable of oxidizing DHA, such as soluble epoxide hydrolase, 12/15-lipoxygenase, or cytochrome P450 epoxygenase, have been

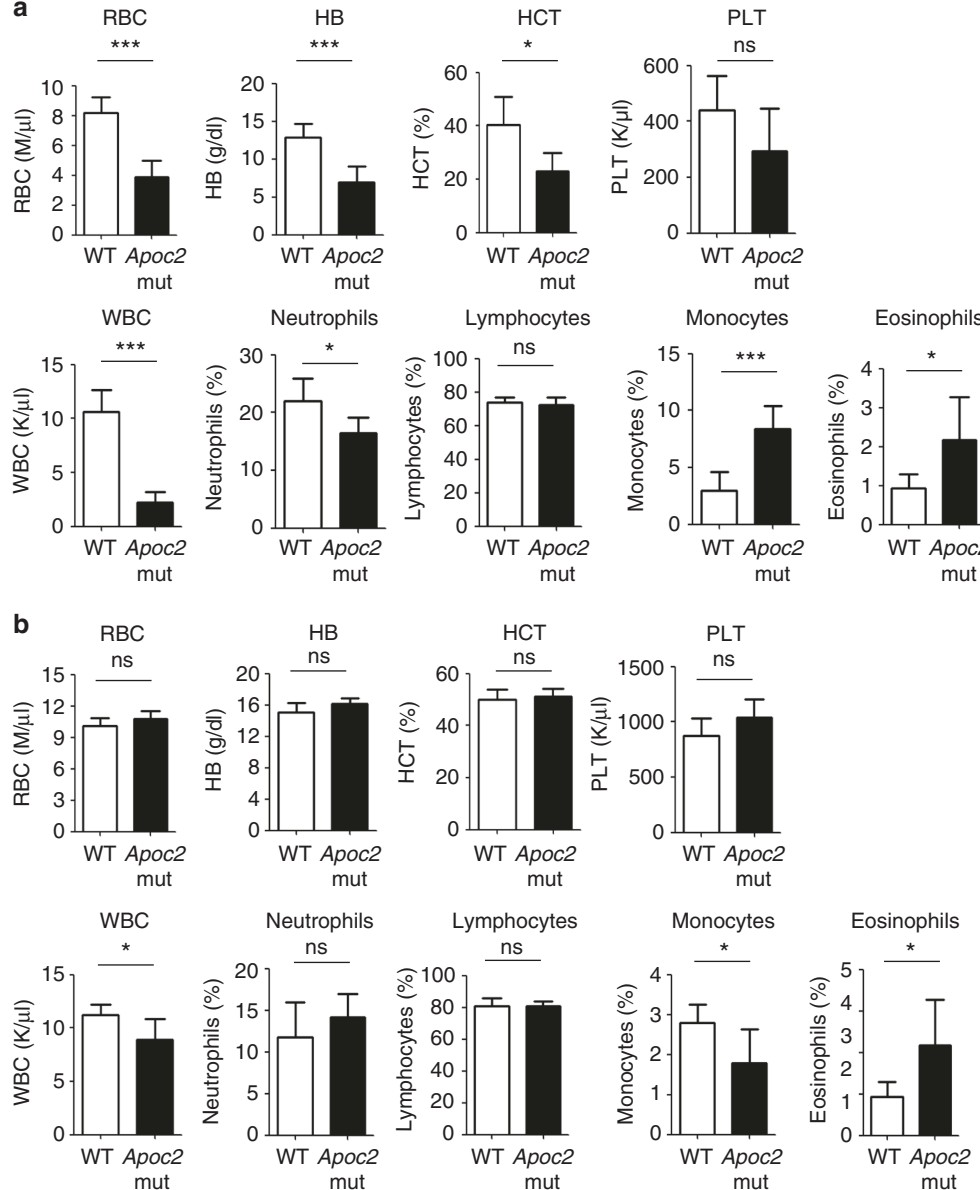

**Fig. 9** Complete blood count in WT and *Apoc2* mutant mice. **a** Blood samples from WT mice and *Apoc2* mutant mice were analyzed at the time of weaning (28–31 days old). WT, *n* = 7 (four males and three females); *Apoc2* mutants, *n* = 5 (two males and three females). No apparent differences between sexes were observed in each group. **b** Blood samples from WT mice and *Apoc2* mutant mice were analyzed at the age of 4–5 months. *n* = 5 in each group, all males. Results are mean ± SEM; *P < 0.05 and ***P < 0.001 (Student's *t* test). WBC white blood cells, RBC red blood cells, HB hemoglobin, HCT hematocrit, PLT platelets. Absolute numbers of WBC and percentages (%) of neutrophils, lymphocytes, monocytes and eosinophils are shown in lower row of graphs in **a** and **b**

reported to regulate HSPCs function[43–45]. Enzymatic products of FFA arachidonic acid (20:4n6) oxidation, prostaglandin E2 (PGE2), and epoxyeicosatrienoic acids, have been shown to regulate HSC homing and engraftment[45, 46]. Clinical applications for PGE2 in umbilical cord blood HSC transplantation have been suggested[47, 48]. Feeding mice with a mix of n6 and n3 PUFAs, including DHA, stimulated hematopoiesis and thrombopoiesis, as well as engraftment of donor cells[49].

In summary, our studies uncovered an important role of LPL activity in regulation of HSPC maintenance and definitive hematopoiesis. The mechanism includes LPL-mediated release of the essential fatty acid DHA to support HSPC maintenance. These findings may have important therapeutic implications, both in terms of dietary recommendations and in optimizing conditions for ex vivo HSPC expansion.

## Methods

**Ethics statement**. All animal experiments were performed according to the NIH guidelines and were approved by the University of California, San Diego Institutional Animal Care and Use Committee (protocols S07266 and S04155). Human plasma, used for VLDL and LDL isolation, was obtained from normal volunteers who provided written informed consent according to a protocol approved by the UC San Diego Human Research Protection Program (project #71402).

**Zebrafish and mouse maintenance**. Adult zebrafish, wild-type (AB strain) and *apoc2* and *lpl* mutants (on the AB background), were maintained at 28 °C, 14-h-light/10-h-dark cycle and fed brine shrimp twice a day. The low-fat diet (LFD) was prepared by extracting lipid from zebrafish GP100–200 micron larval diet (Brine Shrimp Direct, Utah) with diethyl ether. Zebrafish embryos or larvae younger than 5 days post fertilization (dpf) were kept at 28.5 °C in E3 solution (5 mM NaCl, 0.17 mM KCl, 0.33 mM CaCl₂, 0.33 mM MgSO₄). The pigment development of early larvae was inhibited by adding 0.003% *N*-phenylthiourea (Sigma, Cat. 222909) to the E3 medium. *apoc2* mutants[16] and *lpl* mutants zebrafish were generated in our

lab. *cd41*:EGFP and *ahmc*:EGFP transgenic zebrafish were kindly provided by David Traver and Neil Chi (both at UCSD), respectively. *Apoc2* mutant mice were created in Remaley's lab[13] and were fed regular chow diet after weaning. All animal studies were approved by the University of California, San Diego Institutional Animal Care and Use Committee.

**Wright–Giemsa and o-dianisidine staining**. Peripheral blood cells were collected from euthanized zebrafish by tail amputation in adults and by heart puncture in embryos or larvae. Blood smears were made on superfrost/plus slides (Fisher, Cat.12-550-15) and fixed in 100% methanol for 15 s. To visualize erythrocyte morphology, cells were stained using a Hema3 kit (Fisher Diagnostics, Cat. 123–869, an analog to Wright–Giemsa method). To stain for hemoglobin, cells were incubated with an o-dianisidine staining buffer (0.6 mg/ml o-dianisidine, 0.01 M sodium acetate, pH 4.5, 0.65% $H_2O_2$, and 40% ethanol (vol/vol)) for 15 min in the dark. The slides were imaged with a BZ9000 Keyence microscope. For the whole mount o-dianisidine staining, dechorionated and euthanized embryos or larvae were incubated with the o-diansidine staining buffer for 15 min in the dark, and the embryos or larvae were photographed with a Leica CTR5000 microscope. The images shown in Figs. 1 and 2 are representative results from at least two independent experiments, with at least three embryos or larvae in each group per experiment.

**Peripheral blood cell count**. To count blood cells, anesthetized zebrafish embryos or larvae were laterally mounted in 1.0% low-melting agarose in a 50-mm glass bottom dishes (MatTek, Cat. P50G-0-14-F) and the blood flow video were recorded using a BZ9000 Keyence microscope. Three frames were extracted from the video and blood cells were counted manually in each frame, the average number was used for statistical analysis. Blood was collected from adult male zebrafish through tail amputation and peripheral blood cells were counted using a haemocytometer after 1:2000 dilution in PBS.

**In situ hybridization probe synthesis**. Digoxigenin (DIG)-labeled oligonucleotides were synthesized using an in vitro transcription system (Roche, Cat. 11175025910). *gata1*, *cmyb*, *runx1*, and *rag1* probes were synthesized from linearized plasmids with T7 polymerase. *beta-globin*, *pu.1*, and *foxn* probes were synthesized from PCR templates with SP6 polymerase. The PCR templates were amplified from cDNA with specific primers, which were 5′-cgttgctgtcgttctgttta-3′ and 5′-*gatttaggtgacactatag*ttagtggtactgtcttccca-3′ for *beta-globin* (beta embryonic 1.1), 5′-atctatcgcaccaatgga-3′ and 5′-*gatttaggtgacactatagg*cgaagtgttaatgcaaag-3′ for *pu.1* and 5′-agtgtagatggaagtcctgt-3′ and 5′-*gatttaggtgacactatag*ttctccaccttctcaaagca-3′ for *foxn*. The sequence in *italic* encodes the SP6 promoter.

**Whole-mount in situ hybridization**. Whole-mount in situ hybridization (WISH) was performed as described[50]. In brief, embryos were fixed in 4% paraformaldehyde (PFA) overnight at 4 °C and dehydrated sequentially with methanol in PBST (25, 50, and 75%) and stored in 100% methanol at −20 °C. On the day of WISH, embryos were rehydrated in PBST (0.1% Tween-20 in PBS), treated with proteinase K and re-fixed in 4% PFA. After a wash in PBST, the embryos were incubated with hybridization buffer (HB), which contains 50% formamide, 5× saline sodium citrate (SSC), 500 µg/ml torula yeast tRNA, 50 µg/ml heparin, 0.1% Tween-20, and 9 mM citric acid (pH 6.5) for 1 h and then with HB containing DIG-labeled probes overnight at 68 °C. Afterward, embryos were washed sequentially with 2× SSC in HB (25, 50, and 75%) and 0.2× SSC at 68 °C, and then with 0.2× SSC in PBST (75, 50, and 25%) at room temperature (RT). After the wash, embryos were incubated in the blocking buffer (PBST with 2% heat-inactivated goat serum and 2 mg/ml bovine serum albumin) for 1 h at RT and then with blocking buffer containing alkaline phosphatase (AP)-conjugated anti-DIG antibody (Roche, Cat. 11093274910, 1:5000 dilution) overnight at 4 °C. To visualize the signals, embryos were washed six times with PBST for 15 min and then three times with an AP reaction buffer (100 mM Tris, pH 9.5, 50 mM $MgCl_2$, 100 mM NaCl, 0.1% Tween-20) for 5 min at RT. The signals were developed by incubating the embryos with a BM purple AP substrate (Roche, Cat. 11442074001) or an AP reaction buffer containing NBT/BCIP substrate (Roche, 11681451001). The reaction was terminated using a stop buffer (1× PBS, pH 5.2, 1 mM EDTA, 0.1% Tween-20) and the embryos were photographed with a Leica CTR5000 microscope. In situ results shown in Fig. 2d are representative data from three independent experiments. The results in other figures are representative data from two to five independent experiments. The images shown in figures are from the same experiment and the embryo or larva numbers (presented phenotype/total) in each group are indicated in the panels.

**Live imaging**. Anesthetized zebrafish embryos were mounted in 1% low-melting point agarose (Fisher, BP1360-100) containing 0.02% tricaine (Sigma, Cat. A5040) and imaged using a BZ9000 Keyence fluorescent microscope.

**Embryonic injections**. The human APOC2 mimetic peptides C-II-a and C-II-I were reported in previous work[16, 51]. Six nl of C-II-a or C-II-i (2 mg/ml), 8 nl of

VLDL (2.1 mg/ml) or LDL (2.9 mg/ml), 10 nl of DHA (docosahexaenoic acid):BSA (bovine serum albumin) (3 mg/ml:100 mg/ml) or OA (oleic acid):BSA (3.3 mg/ml:129 mg/ml), or 10 nl of POPC:TG-DHA (2 mg/m l:1.2 mg/ml) or POPC (2 mg/ml) liposome were injected into 2 dpf stage embryos through the *sinus venosus* using a FemtoJet micro-injector (Eppendorf).

**Oil red O staining and BODIPY staining**. Oil red O (ORO) staining was conducted according to published protocols[16]. Briefly, embryos were fixed in 4% PFA for 2 h, washed three times in PBS, incubated in 0.3% ORO solution for 2 h, and then washed with PBS before imaging. For BODIPY staining, live larvae were immersed in E3 medium containing 0.1 µg/ml BODIPY 505/515 (Invitrogen, Cat. D-3921) for 1 h in dark and then rinsed with E3 medium before imaging. The images shown in figures are representative results from two to five independent experiments, with at least three embryos or larvae in each group per experiment.

**CRISPR-Cas9-mediated *Lpl* knockout in zebrafish**. pT3TS-zCas9 and T7-gRNA plasmids were from Chen lab[52] through Addgene. Following the published protocol[52], nls-zCas9-nls mRNA was synthesized with an mMESSAGE mMACHINE T3 kit (ThermoFisher, AM1348) and recovered with lithium chloride precipitation. *lpl* gRNA was generated using a MEGAshortscript T7 kit (ThermoFisher, AM1354) and purified using a mirVana miRNA isolation kit (ThermoFisher, AM1560). The zebrafish *lpl* genomic target sequence was 5′-ggctgaaattgattatccttGGG-3′, in which the first 20 nt was the gRNA template and the last 3 nt was protospacer adjacent motif (PAM) required for CRISPR/Cas9 function. 30 pg *lpl* gRNA and 150 pg nls-zCas9-nls mRNA were injected into 1–2 cells stage embryos. Genomic DNA (gDNA) was extracted from whole embryos or from adult tail tissue using a KAPA Express Extract Kit (KAPA Biosystems, Cat. KR0383). The gDNA fragment containing the target site was amplified using KOD DNA polymerase (EMD Millipore, Cat. 71086) and digested with T7 endonuclease (NEB, Cat. M0302). Primers used for PCR amplification of *lpl* gDNA fragment were 5′-aacatcagcctcctacacaa-3′ and 5′-tcactcgtttctcatgcgaa-3′.

**Quantitative RT-PCR**. RNA was isolated from 5 dpf zebrafish embryos using an RNeasy kit (Qiagen, Cat. 74104) and cDNA was reverse transcribed using an EcoRry Premix (Takara-Clontech, Cat. 639543). Quantitative PCR (Kapa SYBR FAST qPCR kit, Cat. KK4602) was performed using a Rotor Gene Q qPCR machine (Qiagen). Primers used in qRT-PCR were 5′-ggcttctgctctgtatgg-3′ and 5′-ggctctgaccttgttgat-3′ for zebrafish *β-actin*, 5′- atgaacaagatactggctat -3′ and 5′-ttgatggtctctacatatcc-3′ for zebrafish *apoc2*, 5′-gcacggcagttcattcaa-3′ and 5′- gtca-gattctaccattccagtt-3′ for zebrafish *lpl*, 5′-cgtcttcacaaaccctcctcaa-3′ and 5′-gctttactgcttcatccggct-3′ for zebrafish *runx1*, and 5′-ccaacagcagcaggtctaa-3′ and 5′-tggtggtctggtggtctt-3′ for zebrafish *sdf1α*.

**Generation of parabiotic zebrafish embryos**. The parabiosis experiment followed the protocol published in ref. [25]. Briefly, embryos for parabiosis were transferred to a glass Petri dish and dechorionated with forceps at the 256-cell stage. Drops of 4% methylcellulose were laid in rows at the bottom of a plastic Petri dish and covered with HCR (116M NaCl, 2.9 mM KCl, 10 mM $CaCl_2$, and 5 mM Hepes) containing antibiotics (2.5 µg/ml ampicillin, 0.5 µg/ml kanamycin, and 10 U/ml penicillin–streptomycin). Small wells were made on top of the methylcellulose drops using the tip of a glass pipette. Then, two individual dechorionated embryos, which developed to the sphere stage but no later than the dome stage, were transferred to these small wells and gently pressed together with a tiny round-end iron needle. At the attaching site of two embryos, a few cells were removed using a sharp glass micropipette. If necessary, the two embryos were moved again to press the wounds against each other. To avoid any shaking that could separate the two blastulae, the attached embryos were left under the microscope for 20–30 min. Once the attachment was secured, the methylcellulose around the embryos was removed as much as possible. Finally, the plates were fully filled with HCR containing antibiotics and transferred to a 28.5 °C incubator. Next day, HCR medium was replaced with E3 medium containing antibiotics and residual methylcellulose was removed.

**Drug treatment**. Lomitapide (Cayman Chem, Cat. 10009610) powders were dissolved in DMSO at the concentration of 1 mM and embryos were treated with 5 µM lomitapide from 2 dpf to 3.3 dpf. DMSO treatment was used as control.

**Gas chromatography – mass spectrometry (GC-MS)**. Pooled embryos or larvae at the dome, 26 h post fertilization (hpf), or 6 dpf stages were homogenated in PBS (10 µl per embryo or larva). Fifty µl of the homogenate was used for free fatty acid extraction, and free fatty acid GC-MS was conducted at the LIPID MAPS Lipidomics Core at UC San Diego[53, 54].

**Apoptosis assay**. Zebrafish embryos (3.3 dpf) were fixed in 4% PFA overnight at 4 °C. Embryos were digested with proteinase K (10 µg/ml) for 30 min and then re-fixed with 4% PFA at RT for 20 min. After the PBST wash, the embryos were incubated with TUNEL reaction mixture (In Situ Cell Death Detection Kit, TMRed, Roche, Cat.12156792910) at 37 °C for 1 h. Positive control embryos were

digested with Dnase I for 15 min. Negative control embryos were incubated with Label solution without terminal transferase. After the PBST wash, embryos were imaged with a BZ9000 Keyence fluorescent microscope.

**Preparation of injected materials.** C-II-a and C-II-i mimetic peptides[51] were dissolved in PBS to a concentration of 2 mg/ml; VLDL (2.1 mg/ml) and LDL (2.9 mg/ml) were from healthy donors and isolated using a standard ultracentrifugation protocol in UC San Diego lipid core. DHA:BSA were made by dissolving 12 mg DHA (Cayman, Cat. 90310), which was first dried under argon, in 4 ml of 100 mg/ml BSA solution (Sigma, Cat. A8806). Oleic acid:BSA was from Sigma (Cat. O3008). Fatty acid concentrations were measured using an HR Series NEFA-HR method (Wako, Cat. 999–34691, 991–34891, 993–35191) and BSA was measured using Lowry assay (Biorad, Cat. 500–0116). To make POPC:TG-DHA and POPC liposomes, chloroform solubilized POPC (Avanti, Cat. 850457) and tridocosa-hexaenoin (TG-DHA) (Larodan Cat. 33–2260) were mixed at 2 mg POPC with 1.2 mg TG-DHA or 2 mg POPC alone. The mixtures were dried under argon in round bottom glass tubes and then 1 ml PBS was added and incubated at RT for 30 min. Then, POPC:TG-DHA or POPC in PBS was sonicated in a water bath ultrasound machine (VWR, Model 75D) three times for 10 min, until the solution became uniform and translucent.

**Mouse complete blood count.** Blood samples (~50 μl) were collected into EDTA tubes (BD, Cat. 365974) from mouse tail. To minimize clotting, the tubes were flicked immediately after blood collection and the samples were sent to the UCSD Veterinary Diagnostic Laboratory for CBC analysis within 2 h.

**FACS sorting of HSPCs and stromal cells.** EGFP and mCherry double-positive embryos were picked up from progenies of the cross between *gata2*:EGFP and *sdf1a*:mCherry zebrafish at 2.5 dpf. Caudal hematopoietic tissues (CHT) were cut out and washed with PBS. CHT from about 200 embryos were digested with 1 ml Trypsin-EDA (Corning, Cat. 25-052-CI) containing 10 μg/ml collagenase (Worthington Biochem, Cat. MOP123) at 35 °C. FBS (Omega Science, Cat. FB-01) was added to a final concentration of 5% to stop the reaction. Cells were spun down at 350 × *g* for 5 min, re-suspended in PBS and stained with Aqua-LIVE/DEAD kit (ThermoFisher, Cat. L34965). Cells were spun down, washed once with PBS, and re-suspended in 0.5 ml staining buffer (PBS containing 0.5% BSA). After filtration through a 70 μm cartridge, cells were sorted using a BD FACSAria II machine. Live EGFP single-positive cells, mCherry single-positive cells, and EGFP/mCherry double-negative cells were defined as HSPCs, stromal cells, and trunk cells, respectively.

Bone marrow cells were collected from mouse femur and tibia and depleted from red blood cells with a red blood cell lysis buffer (eBioscience, Cat. 00–4333). Cells were spun down at 350 × *g* for 5 min, re-suspended in the staining buffer to 10 million per ml, and stained with Lin-APC (BD, Cat. 55807, 1:1000 dilution), cKit-PE (eBioscience, Cat. 12-1172-82, 1:300 dilution), and Sca1-PEcy7 (eBioscience, Cat. 25-5981-81, 1:2500 dilution) antibodies on ice for 15 min. Cells were spun down and washed once with PBS and re-suspended in the staining buffer to 3–5 million per ml. Cells were sorted using a BD FACSAria II machine and fluorescence minus one (FMO) and single stain controls were used to calibrate the threshold in the FACS sorting. Cells with lineage negative staining (Lin$^{-}$) were further selected for positive Sca1 and c-Kit staining (Sca1$^{+}$ and c-Kit$^{+}$), and these cells were defined as LSK.

**Data availability.** The authors declare that all data supporting the findings of this study are available within the article and its supplementary information files or from the corresponding author on reasonable request.

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

## Acknowledgements
This study was supported by grants HL135737, HL136275, and HL088093 (Y.I.M.) from the National Institutes of Health and 16POST27250126 (C.L.) from the American Heart Association. The UCSD School of Medicine Microscopy Core is supported by grant P30 NS047101 from the NIH. Equipment is supported by Jennifer Santini.

## Author contributions
Studies were designed and planned by C.L., Y.I.M, and D.T.; experiments and data analysis were performed by C.L., T.H., D.L.S., H.W., B.L.V., and J.K; C.L. and Y.I.M. wrote the manuscript; R.L.K., A.T.R, T.M.R., and D.T. contributed to study discussions and manuscript revisions.

## Additional information

**Competing interests:** The authors declare no competing interests.

