## [Peer Review File · Nature Communications]

Reviewers' comments:

Reviewer #1 (Remarks to the Author):

Reviewer comments:

"Lipoprotein lipase regulates hematopoietic stem progenitor cell maintenance through DHA supply"

General comments:

The authors characterize for the first time the unexpected role of lipolysis in definitive hematopoiesis. Appropriate controls are included to conclusively show that hyperlipidemia does not inhibit hematopoiesis per se, but rather free fatty acids liberated through lipolysis are required to promote the process. Mass spectrometry experiments highlight a small pool of fatty acids that may be responsible for this phenotype, and rescue experiments show that free DHA is able to rescue several aspects of the hematopoiesis defects observed in mutants. These findings represent an important advance in the field that enhances our fundamental understanding of vertebrate biology and could lead to translational outcomes that improve human health. The experiments presented make a compelling case that will be of interest to readers Nature Communications. In short, these results are novel and important. That said, there are several important issues must be addressed prior to publication.

Major issues:

1) A major issue with this publication is that the parabiosis experiments (figure 5) and DHA supplementation (figure 6) are only shown to increase expression of hematopoietic markers (using in situ hybridization). It is essential that the authors report whether (and to what degree) these gene expression changes ultimately result in amelioration of anemia using red blood cell counts/characterization (as performed in figure 1). This is particularly important given that in situ hybridizations are not a quantitative measure of gene expression, meaning there is no quantitative evidence supporting several of the key findings of the manuscript.

2) A second major issue with the manuscript is that it fails to cite important previous work describing the role of lipids and lipid derivatives in hematopoiesis and HSPC maintenance, much of which has been carried out in zebrafish. For example, PGE2 is derivative from a polyunsaturated fatty acid, expected to be produced by lipolysis and promotes HSC maintenance (PMID: 17581586), and is currently undergoing clinical trials to expand HSCs, which is stated as a potential translational outcome of this manuscript as well. A non-comprehensive literature search also revealed numerous prior publications demonstrating the importance of DHA (PMID: 20230224) or unsaturated fatty acids (PMID: 28570944) in hematopoiesis. Although this work appears to provide the first evidence for the necessity of free/lipolyzed circulating DHA in this process, it is essential that these findings be discussed in the context of previous literature directly relevant to these findings. Additional topics that may warrant discussion are (i) if phospholipid lipolysis is perturbed or important for this process, and (ii) whether alternative unsaturated fatty acids could have similar effects. I would also have expected that there would have been some mention of work from the Weinstein and Yaniv labs that explored the role of LDL on vessel growth by manipulating some of the very same lipolysis pathways (apoC2 and MTP) (PMID: 22581286).

3) A pervasive issue throughout the paper was that it was not clear how many experiments have been performed. Many figures (such as 1A, 1E, 2A, 2D, ...) lack any indication of the number of biological replicates or number of experiments performed. Even in cases where numbers of biological replicates

are clearly indicated (such as figure 3A), it is not clear whether these results were all from the same experiment. It is standard practice in the field to perform at least 2 (if not 3) independent experiments with several biological replicates per experiment and report these numbers in the methods or figure legends.

4) It struck me as odd that figures reporting data from mouse experiments are exclusively in the supplemental material. These data are important for the interpretation of the results and conclusions and should be accessible in the main body of the text.

5) The *lpl* allele is discussed as resulting in a truncated protein product, but there is emerging evidence that many mechanisms exist that can result in different mRNA and protein products than would be predicted. For example, a frameshift/early termination mutation can lead to in-frame exon skipping (PMID: 28570605) or use of alternative start codons (PMID: 28280001). Support for the generation of truncated protein production in mutants would require the use of a *lpl* antibody, but if such an antibody is not available then sequencing of the cDNA or at minimum discussion of the intron/exon structure and discussion of the recent literature surrounding this topic would enable more accurate prediction of protein products.

6) Figure S7 is used to conclude that there is no somatic cell exchange between parabiotic animals. However, the transgenic line used expresses a fluorescent marker only in muscle tissue, so the only conclusion that can be made is that there is not transfer of muscle cells from one individual to the other. Even this conclusion is quite tenuous, as the image has an unacceptably high level of background fluorescence. If the authors would like to conclusively state that there is no somatic cell transfer between parabiotic individuals, these experiments must be repeated using zebrafish that express a ubiquitous marker, and the resulting image must be taken with sufficient quality/low enough background that it is possible to differentiate between background signal and somatic cell transfer. If this is not possible, the authors must honestly state that they cannot eliminate the possibility of somatic cell transfer, although other experiments do provide support that somatic cell transfer is not likely responsible for the rescue observed.

Minor issues:

-The low-magnification images in figures 1E and 2L should be removed and the insets should be enlarged (~4 fold) so that the image can be seen and interpreted clearly.

-The discussion of figure 3C-3F should be clarified. It must be clearly stated that there appear to be two distinct defects in the *apoc2* mutants, the first being that fewer HSPCs are present, and the second being that even though differentiation into thrombocytes appears normal, there appears to be a defect in their subsequent migration to the thymus.

-A citation should be added to support the expression and ramifications of CETP expression in zebrafish (PMID:21855599).

-A citation should be added demonstrating the effective previous use of lomitapide (MTP inhibitor) in zebrafish (PMID: 27655916).

-Discussion should be expanded on the ability of mice to recover from anemia later in life. Most importantly, it should be discussed whether the *apoc2* mutant (3 AA deletion) is hypomorphic or a total loss of function, or whether alternative lipases such as phospholipases, may contribute to the DHA pool and recovery of anemia.

Reviewer #3 (Remarks to the Author):

This well written and elegantly documented paper examines the effects of LpL and ApoC2 deficiency in a Zebrafish model. The paper tells a nice story regarding the need for TG hydrolysis to supply DHA, which is apparently needed for hematopoietic stem cell development. There are a few comments, which are relatively minor.

An obvious potential criticism surrounds the relevance of these findings to mammals, especially humans. The authors should dig into the literature to better understand the manifestations of LpL deficiency in infancy. One clinical paper (Feoli-Fonseca et al, J. Pediatrics, PMID9738727) describes considerable anemia in infants with LpL deficiency (although they sometimes get GI bleeds and have other problems), which seems to correct over time. Perhaps like the mice described in Fig S11.

The rescue of the fish with DHA as a FFA is interesting. Presumably zebrafish do not have the ability to synthesize DHA from ALA or other precursors. If this is known, it would be useful to state. It would be of interest to know if this requirement is absolute for DHA, or whether other n-3 PUFAs could serve this function. Some speculation about what is it about DHA that supports stem cell maintenance would be useful.

Fig S11 indicates some changes that are not explained. In the ApoC2 mutants, eosinophils and monocytes are increased, in the face of all other cell populations being low. Also, the WBC population is decreased by 5-10x, yet the changes in the neutrophils, lymphocytes, etc do not match. The same apparent discrepancy holds for the 4-5 month old mice.

Reviewer #4 (Remarks to the Author):

In this work, Liu et al. have shown that loss of function mutations in the *lpl* and *apoc2* genes cause the deficiency of definitive haematopoiesis and anemia in zebrafish. The authors went on to show that the haematopoietic defects in *lpl* and *apoc2* mutant fish can be rescued by simply injecting FFA-DHA into the circulation of the mutant animals. Based on these data, the author concluded that lipid metabolism is critical for regulating HSPC development. Although this study is interesting, the current results are preliminary and lack of in-depth understanding of the underlying mechanism, which are prerequisite for its publication in Nature Communications.

Major concerns:

1. Previous study by Liu et al. showed that *apoc2* mutant fish have smaller body size and impaired vascular development at early stage (Liu et al., 2015). This suggests that the haematopoietic defect in these mutant fish, at least in *apoc2* mutants, is not haematopoiesis specific but rather is a consequence of an overall developmental delay caused by fatty acid metabolic deficiency. If so, their findings will be less significance. The authors should perform necessary experiments to exclude this possibility, which is critical for the publication of their work.
2. Despite characterization of the mutant haematopoietic phenotypes and the DHA rescue experiment, it remains unclear how DHA executes its function in this process at the cellular and molecular levels. These issues need to be clarified.

Minor concerns:

1. Lpl is a well-known lipoprotein lipase that catalyzes hydrolysis of triglycerides to supply many kinds of free fatty acids, including DHA. It will be interesting to test whether other metabolites catalyzed by Lpl/Apoc2 can also rescue the haematopoietic defect in lpl and apoc2 mutants.

2. The authors showed that in adult mice bone marrow, stromal cells are the main resource of Lpl while haematopoietic cells rarely express Lpl. However, the adult haematopoietic niche is clearly different from the embryonic niche. Thus, the authors should examine which tissues or cell populations in zebrafish embryos are responsible for producing Lpl and Apoc2.

3. In the main text, after describing Figure 5, the author drew the conclusion that '.....The Apoc2/Lpl-mediated release of FFAs into shared circulation then rescues haematopoiesis in both mutants, even though the Lpl deficiency persists in the CHT niche in lpl mutants.' However, they didn't provide any direct evidence to support such conclusion until Figure 6. This is an obvious logic error and should be corrected.

Dear Editor,

Thank you for reviewing our work and for your guidance on how to revise the manuscript. We are pleased that the Reviewers found our work of considerable potential interest and appreciate many thoughtful comments, which were very helpful in revising and strengthening the paper.

As we discussed, we were able to address all Reviewers' concerns, many with new experimental results supporting conclusions of our manuscript. As you have pointed out, among other changes, we carefully addressed Reviewer #3's concern of an overall developmental delay in *apoc2* mutants. We now present strong data showing that a slight delay in developmental angiogenesis is not the cause of the hematopoietic phenotype in the mutants.

Addressing the question of how DHA may exert its effects on HSPC maintenance, we discussed our findings in the context of previous literature directly relevant to role of DHA in hematopoiesis, as Reviewer #2 suggested. However, we feel that the experimental identification of the exact molecular mechanism by which free fatty acid DHA executes its function in zebrafish hematopoiesis is well beyond the scope of the current work and will constitute a large, separate study. This work cannot be reasonably completed as a revision of the current paper, which is already large (64 figure panels and 2 movies).

The novelty of our findings is in the discovery of the role of triglyceride lipolysis in hematopoiesis and the realization that DHA must be in a free fatty acid form to exert its effect on HSPC maintenance. We believe these results, now strengthened with important controls and additional experiments requested by the Reviewers, significantly contribute to the current understanding of hematopoiesis and will be of considerable interest for the Nature Communications readership.

Below is the point-by-point response to Reviewers' comments.

Thank you again for considering our manuscript and we appreciate your guidance on the required revision.

Best regards,

Yury Miller

Reviewer #1

General comments:

The authors characterize for the first time the unexpected role of lipolysis in definitive hematopoiesis. Appropriate controls are included to conclusively show that hyperlipidemia does not inhibit hematopoiesis per se, but rather free fatty acids liberated through lipolysis are required to promote the process. Mass spectrometry experiments highlight a small pool of fatty acids that may be responsible for this phenotype, and rescue experiments show that free DHA is able to rescue several aspects of the hematopoiesis defects observed in mutants. These findings represent an important

advance in the field that enhances our fundamental understanding of vertebrate biology and could lead to translational outcomes that improve human health. The experiments presented make a compelling case that will be of interest to readers *Nature Communications*. In short, these results are novel and important.

Response: We appreciate Reviewer's positive evaluation of the novelty and importance of our work.

That said, there are several important issues must be addressed prior to publication.

Major issues:

1) A major issue with this publication is that the parabiosis experiments (figure 5) and DHA supplementation (figure 6) are only shown to increase expression of hematopoietic markers (using *in situ* hybridization). It is essential that the authors report whether (and to what degree) these gene expression changes ultimately result in amelioration of anemia using red blood cell counts/characterization (as performed in figure 1). This is particularly important given that *in situ* hybridizations are not a quantitative measure of gene expression, meaning there is no quantitative evidence supporting several of the key findings of the manuscript.

Response: We agree, this is an important point, and circulating cell count is an ultimate measure of anemia. In the DHA rescue experiments, we now counted circulating cells at 6.3 dpf. Consistent with the *in situ* results, DHA, but not OA (oleic acid) or EPA (eicosapentaenoic acid; added to this figure in response to other Reviewer's comments), significantly increased cell numbers. In addition, we found that DHA, but not OA or EPA, reduced the hypochromia in red blood cells and increased hemoglobin staining in *apoc2* mutants at 6.3 dpf. New data are shown below and as Fig. S9 in the manuscript:

Zebrafish parabiosis remains to be a challenging experimental technique and in our hands, parabiotic zebrafish begin to develop abnormally beyond 5 dpf, displaying cardiac edema and unequal/asymmetric development and dying shortly thereafter. We were unable to reliably count blood cells in parabiotic zebrafish. We did not find any references to work in which parabiotic zebrafish were maintained to 6 dpf, the time point at which we detected reduced blood cell counts.

2) A second major issue with the manuscript is that it fails to cite important previous work describing the role of lipids and lipid derivatives in hematopoiesis and HSPC maintenance, much of which has been carried out in zebrafish. For example, PGE2 is derivative from a polyunsaturated fatty acid, expected to be produced by lipolysis and promotes HSC maintenance (PMID: 17581586), and is currently undergoing clinical trials to expand HSCs, which is stated as a potential translational outcome of this manuscript as well. A non-comprehensive literature search also revealed numerous prior publications demonstrating the importance of DHA (PMID: 20230224) or unsaturated fatty acids (PMID: 28570944) in hematopoiesis. Although this work appears to provide the first evidence for the necessity of free/lipolyzed circulating DHA in this process, it is essential that these findings be discussed in the context of previous literature directly relevant to these findings.

Additional topics that may warrant discussion are (i) if phospholipid lipolysis is perturbed or important for this process, and (ii) whether alternative unsaturated fatty acids could have similar effects. I would also have expected that there would have been some mention of work from the Weinstein and Yaniv labs that explored the role of LDL on vessel growth by manipulating some of the very same lipolysis pathways (apoC2 and MTP) (PMID: 22581286).

Response: Thank you very much for these helpful suggestions. We have integrated these and other references and topics into our revised Discussion.

3) A pervasive issue throughout the paper was that it was not clear how many experiments have been performed. Many figures (such as 1A, 1E, 2A, 2D, ...) lack any indication of the number of biological replicates or number of experiments performed. Even in cases where numbers of biological replicates are clearly indicated (such as figure 3A), it is not clear whether these results were all from the same experiment. It is standard practice in the field to perform at least 2 (if not 3) independent experiments with several biological replicates per experiment and report these numbers in the methods or figure legends.

Response: We have revised the Methods section to indicate how many independent experiments were performed for each type of assay. There was never only a single experiment performed. From two to five independent experiments, with multiple biological replicates in each, were performed for each figure panel in this manuscript.

4) It struck me as odd that figures reporting data from mouse experiments are exclusively in the supplemental material. These data are important for the interpretation of the results and conclusions and should be accessible in the main body of the text.

Response: We have moved the mouse data to the main body of the manuscript. It is now Fig. 7.

5) The *lpl* allele is discussed as resulting in a truncated protein product, but there is emerging evidence that many mechanisms exist that can result in different mRNA and protein products than would be predicted. For example, a frameshift/early termination mutation can lead to in-frame exon skipping (PMID: 28570605) or use of alternative start codons (PMID: 28280001). Support for the generation of truncated protein production in mutants would require the use of a *lpl* antibody, but if such an antibody is not available then sequencing of the cDNA or at minimum discussion of the intron/exon structure and discussion of the recent literature surrounding this topic would enable more accurate prediction of protein products.

Response: Thank you for raising this issue. It is an important control we had to perform. We have tried all the available LPL antibodies on the market and also the LPL antibody obtained from Dr. Andre Bensadoun (Cornell University). However, none of them could detect zebrafish *Lpl*.

To test in-frame exon skipping or alternative splicing, we designed primers spanning exon 2 and exon 10 to flank the exon 4, the site of the CRISPR generated mutation. There was only 1 single cDNA band of the same size in WT and *lpl* mutants. Sequencing the cDNA confirmed that the only *lpl* mutant cDNA contained the 2nt deletion, same as the result from genomic DNA sequence. The alternative start codon usually requires a short ORFs akin to uORFs (upstream open reading frames, the majority of which is less than 20 amino acid). However, our 2nt deletion occurred in exon 4 and the resulting ORF (192 aa) is much larger. This is unlikely to cause a re-initiation of translation and a truncated C-terminal protein. Even if this happens, the resulting protein would lack the N-terminal lipase domain and thus be non-functional. We concluded that our *lpl* mutant is a loss-of-function mutant. And indeed, hypertriglyceridemia similar to that in *apoc2* KO mutants, is observed in *lpl* mutants. New data are in Fig. S3, copied in part here:

6) Figure S7 is used to conclude that there is no somatic cell exchange between parabiotic animals. However, the transgenic line used expresses a fluorescent marker only in muscle tissue, so the only conclusion that can be made is that there is not transfer of muscle cells from one individual to the other. Even this conclusion is quite

tenuous, as the image has an unacceptably high level of background fluorescence. If the authors would like to conclusively state that there is no somatic cell transfer between parabiotic individuals, these experiments must be repeated using zebrafish that express a ubiquitous marker, and the resulting image must be taken with sufficient quality/low enough background that it is possible to differentiate between background signal and somatic cell transfer. If this is not possible, the authors must honestly state that they cannot eliminate the possibility of somatic cell transfer, although other experiments do provide support that somatic cell transfer is not likely responsible for the rescue observed.

Response: Thank you for raising this issue. While addressing this concern we found the way to support with new zebrafish data our findings of LPL expression in mouse stromal cells but not HSPCs!

We sorted out HSPCs and stromal cells from the zebrafish expressing EGFP driven by the *gata2* promoter (HSPCs) and mCherry driven by *sdf1a* promoter (stromal cells). The sorting results were confirmed by testing mRNA expression of *runx1* and *sdf1a* in isolated fractions. Because there were 100-fold more stromal cells than HSPCs and because stromal cells but not HSPCs highly expressed *lpl*, we concluded that stromal cells are the major source of *lpl* expression in the CHT. The results are copied below (panels A-C) and are now in Fig. 5.

Back to the issue of somatic cell exchange in parabiotic zebrafish, in new experiments, we fused a *sdf1a*:mCherry or an *ahmc*:EGFP transgenic embryo with a WT embryo, and found no detectable tissue expression of mCherry or EGFP in the WT embryo. We present better quality images of *ahmc*:EGFP, as requested. More importantly, the *sdf1a*:mCherry-WT parabiosis did not result in transfer of stromal cells (red) from *sdf1a*:mCherry into the CHT region of WT embryos. These data are copied below (panels D and E) and is part of Fig. 5.

Minor issues:

-The low-magnification images in figures 1E and 2L should be removed and the insets should be enlarged (~4 fold) so that the image can be seen and interpreted clearly.

Response: We have replaced images as suggested.

-The discussion of figure 3C-3F should be clarified. It must be clearly stated that there appear to be two distinct defects in the apoc2 mutants, the first being that fewer HSPCs are present, and the second being that even though differentiation into thrombocytes appears normal, there appears to be a defect in their subsequent migration to the thymus.

Response: We have clarified this issue in the text. There appears to be a defect in HSPC migration to thymus where they differentiate into lymphocytes.

-A citation should be added to support the expression and ramifications of CETP expression in zebrafish (PMID:21855599).

Response: We have added the citation.

-A citation should be added demonstrating the effective previous use of lomitapide (MTP inhibitor) in zebrafish (PMID: 27655916).

Response: We have added the citation.

-Discussion should be expanded on the ability of mice to recover from anemia later in life. Most importantly, it should be discussed whether the apoc2 mutant (3 AA deletion) is hypomorphic or a total loss of function, or whether alternative lipases such as phospholipases, may contribute to the DHA pool and recovery of anemia.

Response: Thank you for this suggestion! We have revised Discussion accordingly.

Reviewer #2

This well written and elegantly documented paper examines the effects of LpL and ApoC2 deficiency in a Zebrafish model. The paper tells a nice story regarding the need for TG hydrolysis to supply DHA, which is apparently needed for hematopoietic stem cell development. There are a few comments, which are relatively minor.

Response: Thank you for the complimentary remarks.

An obvious potential criticism surrounds the relevance of these findings to mammals, especially humans. The authors should dig into the literature to better understand the manifestations of LpL deficiency in infancy. One clinical paper (Feoli-Fonseca et al, J. Pediatrics, PMID9738727) describes considerable anemia in infants with LpL deficiency (although they sometimes get GI bleeds and have other problems), which seems to correct over time. Perhaps like the mice described in Fig S11.

Response: We have added this citation and expanded Discussion to discuss the relevance of our findings to mammals.

The rescue of the fish with DHA as a FFA is interesting. Presumably zebrafish do not have the ability to synthesize DHA from ALA or other precursors. If this is known, it would be useful to state. It would be of interest to know if this requirement is absolute for DHA, or whether other n-3 PUFAs could serve this function. Some speculation about what is it about DHA that supports stem cell maintenance would be useful.

Response: Just like other vertebrates, zebrafish have lost the capacity to synthesize n3 PUFA *de novo* due to the lack of delta-12 desaturase and delta-15 desaturase ¹. Although zebrafish may synthesize DHA from dietary ALA, without exogenous over-expressed delta6-desaturase and other enzymes, the process is of low efficiency, as most ALA is used in mitochondria for beta-oxidation ². So, yes, the major source of DHA in zebrafish is from the diet. As we discuss in the manuscript, the dietary FA are re-esterified into TG during intestinal absorption, and our findings suggest the importance of DHA function as a FFA and the intriguing role of LPL mediated hydrolysis in releasing DHA from TG-DHA.

Thank you for the suggestion to test a different n3 PUFA. In new experiments, we injected EPA into *apoc2* mutants and found that in contrast to DHA, it did not rescue the hematopoietic defects. Also, we noticed that EPA and ALA levels were not changed in our GC-MS data. Thus, our data suggest that DHA (or its derivatives) selectively regulates zebrafish hematopoiesis. New, EPA-related data are copied below and in Fig. S9, and new discussion of how DHA may support stem cell maintenance is added to the text.

Fig S11 indicates some changes that are not explained. In the ApoC2 mutants, eosinophils and monocytes are increased, in the face of all other cell populations being low. Also, the WBC population is decreased by 5-10x, yet the changes in the neutrophils, lymphocytes, etc do not match. The same apparent discrepancy holds for the 4-5 month old mice.

Response: Sorry for the confusion. We re-arranged and re-labeled figure panels for better presentation. WBC are in absolute numbers (K/ μ l), and neutrophils, lymphocytes, monocytes and eosinophils are presented as % of total WBC. Fig. S11 is now Fig. 7 in the main body, as suggested by Reviewer #1.

References used in responses:

1. Nakamura MT, Nara TY. Structure, function, and dietary regulation of delta6, delta5, and delta9 desaturases. *Annu Rev Nutr* **24**, 345-376 (2004).
2. Alimuddin, Yoshizaki G, Kiron V, Satoh S, Takeuchi T. Enhancement of EPA and DHA biosynthesis by over-expression of masu salmon delta6-desaturase-like gene in zebrafish. *Transgenic Res* **14**, 159-165 (2005).

Reviewer #3

*In this work, Liu et al. have shown that loss of function mutations in the *lpl* and *apoc2* genes cause the deficiency of definitive haematopoiesis and anemia in zebrafish. The authors went on to show that the haematopoietic defects in *lpl* and *apoc2* mutant fish can be rescued by simply injecting FFA-DHA into the circulation of the mutant animals. Based on these data, the author concluded that lipid metabolism is critical for regulating HSPC development. Although this study is interesting, the current results are preliminary and lack of in-depth understanding of the underlying mechanism, which are prerequisite for its publication in Nature Communications.*

Response: Thank you for finding our study interesting. Below we address your concerns.

Major concerns:

*1. Previous study by Liu et al. showed that *apoc2* mutant fish have smaller body size and impaired vascular development at early stage (Liu et al., 2015). This suggests that the haematopoietic defect in these mutant fish, at least in *apoc2* mutants, is not haematopoiesis specific but rather is a consequence of an overall developmental delay caused by fatty acid metabolic deficiency. If so, their findings will be less significant. The authors should perform necessary experiments to exclude this possibility, which is critical for the publication of their work.*

Response: We agree that addressing the issue of a developmental delay is critical for this work. In our previous work (Liu et al. 2015), we indeed noticed delayed angiogenesis, but we also confirmed that the overall development stage, monitored by the pigmentation pattern and the head angle, is not affected in *apoc2* mutants. The delayed angiogenesis was most obvious at an early stage, but, in this work, we found that the early HSPC

specification was not affected in the ventral aspects of the dorsal aorta (VDA) at 26hpf (Fig. 3A of this manuscript).

To further test that the hematopoietic defect was not due to delayed angiogenesis, we synchronized *apoc2* mutants with WT by maintaining *apoc2* mutant embryos at 30°C from 72 to 80 hpf, while WT embryos were kept at 28°C. As shown below and in new Fig. S5, an angiogenesis stage in WT and synchronized *apoc2* mutants was identical (yellow arrow), as was the size of the yolk (red arrow). Importantly, there remained a low expression of *runx1/cmyb* in the CHT region of the synchronized *apoc2* mutants, indicating that the hematopoietic defect was not due to a slightly delayed angiogenesis in the mutants. As in our earlier experiments, the DHA injection rescued the *runx1/cmyb* expression, but, importantly, had no effect on the angiogenesis progression.

In addition, delayed, not impaired angiogenesis in *apoc2* mutants is fully resolved and naturally synchronizes with WT at later stages. Still, severe anemia phenotypes were observed at 6.3 dpf and in *apoc2* mutant adults. In our other experiments, an APOC2 mimetic peptide and DHA, administered at 2 dpf, rescued the hematopoietic defects, also suggesting that defective lipolysis and reduced DHA levels affect hematopoiesis.

2. Despite characterization of the mutant haematopoietic phenotypes and the DHA rescue experiment, it remains unclear how DHA executes its function in this process at the cellular and molecular levels. These issues need to be clarified.

Response: Addressing the question of how DHA may exert its effects on HSPC maintenance, we modified Discussion to consider our findings in the context of previous literature directly relevant to role of DHA in hematopoiesis, as Reviewer #2 suggested. However, we feel that the experimental identification of the exact molecular mechanism by which free fatty acid DHA executes its function in zebrafish hematopoiesis is beyond the scope of the current work and will constitute a large, separate study. This work cannot be reasonably completed as a revision of the current paper, which is already large (64 figure panels and 2 movies).

The novelty of our findings is in the discovery of the role of triglyceride lipolysis in hematopoiesis and the realization that DHA must be in a free fatty acid form to exert its

effect on HSPC maintenance. We believe these results, now strengthened with important controls and additional experiments requested by the Reviewers, significantly contribute to the current understanding of hematopoiesis and will be of considerable interest for the Nature Communications readership.

Minor concerns:

1. *Lpl* is a well-known lipoprotein lipase that catalyzes hydrolysis of triglycerides to supply many kinds of free fatty acids, including DHA. It will be interesting to test whether other metabolites catalyzed by *Lpl*/*Apoc2* can also rescue the haematopoietic defect in *lpl* and *apoc2* mutants.

Response: Thank you for this suggestion. In our original work, we tested the effect of oleic acid (OA), and in new experiments, we injected eicosapentaenoic acid (EPA, another n3 PUFA) into *apoc2* mutants and found that in contrast to DHA, OA nor EPA rescued the hematopoietic defects. Thus, our data suggest that DHA selectively regulates zebrafish hematopoiesis. New data are copied below and in Fig. S9, and new discussion of how DHA may support stem cell maintenance is added to the text.

2. The authors showed that in adult mice bone marrow, stromal cells are the main resource of *Lpl* while haematopoietic cells rarely express *Lpl*. However, the adult haematopoietic niche is clearly different from the embryonic niche. Thus, the authors should examine which tissues or cell populations in zebrafish embryos are responsible for producing *Lpl* and *Apoc2*.

Response: Thank you for this suggestion! In new experiments, we crossed *gata2:EGFP* and *sdf1a:mCherry* zebrafish and used the double positive embryos to isolate the CHT region and then FACS sort EGFP-positive HSPCs and mCherry-positive stromal cells. The cells were used to perform RT-qPCR to verify cells identity (*runx1* and *sdf1a*) and measure *lpl* expression. Because there were 100-fold more stromal cells than HSPCs and because stromal cells but not HSPCs highly expressed *lpl*, we concluded that stromal cells are the major source of *lpl* expression in zebrafish, consistent with the mouse data. These data are copied below and in new Fig. 5A-C.

Apoc2 is a secreted protein and its expression in zebrafish embryos localizes to the intestine and yolk, as shown in Fig. S2 and below.

3. In the main text, after describing Figure 5, the author drew the conclusion that ‘.....The *Apoc2/Lpl*-mediated release of FFAs into shared circulation then rescues haematopoiesis in both mutants, even though the *Lpl* deficiency persists in the CHT niche in *lpl* mutants.’ However, they didn’t provide any direct evidence to support such conclusion until Figure 6. This is an obvious logic error and should be corrected.

Response: Thank you. We have amended this statement.

REVIEWERS' COMMENTS:

Reviewer #1 (Remarks to the Author):

The authors were extremely responsive to the issues raised in the original review and nicely addressed ALL of the concerns.

Reviewer #4 (Remarks to the Author):

Although the revised manuscript has made some improvement, the following key issues remains unaddressed.

1. Lipid metabolism plays a critical role in embryonic development through affecting various tissues and cell types, including but not limited to vascular system. Although the authors provided data to show that the structure of vessel is relatively normal in *apoc2* mutants, it is insufficient to prove that *apoc2* and FFA-DHA directly affect haematopoiesis. It remains possible that the haematopoietic phenotype is caused by the delay of the whole embryonic development. This possibility is supported by their data showing that the size of *apoc2* mutant embryos is smaller than that of WT embryos (Fig. 1E and F, Fig. 2L and Fig. 3B showed that the body length and head size of *apoc2* mutants are smaller than WT embryos). Indeed, after DHA treatment of *apoc2* mutants, both the haematopoietic and the body size defects were rescued at the same time (Fig. S5 A and S9 B), showing that the general developmental delay and haematopoietic defect cannot be disassociated. To prove the direct function of FFA-DHA in haematopoiesis, the authors should develop a method in which they could supply FFA-DHA specifically into HSCs in *apoc2* mutants to see whether it's sufficient to rescue the haematopoietic defect. A more specific assay will be necessary to clarify this issue.

2. Because lipid metabolism plays a critical role in general development of animals, the current data could not distinguish whether FFA-DHA have a specific role in hematopoiesis. It is thus important to uncover the molecular basis underlying the action of FFA-DHA on haematopoiesis, which, I believe, may provide a definitive evidence demonstrating that FFA-DHA indeed have a specific role in haematopoiesis.

Reviewer #5 (Remarks to the Author):

When dealing with general pathways like lipids, tissue specific effects can be difficult to assess because of embryonic developmental delays. Here the authors have really done a good job of matching the blood vessel timing, so I think the conclusions hold. I think the rescue experiments are quite good here and nailing the DHA as the mechanism.

NCOMMS-17-13519A

Response to reviewers' comments

REVIEWERS' COMMENTS:

Reviewer #1 (Remarks to the Author):

The authors were extremely responsive to the issues raised in the original review and nicely addressed ALL of the concerns.

Reviewer #4 (Remarks to the Author):

Although the revised manuscript has made some improvement, the following key issues remains unaddressed.

1. Lipid metabolism plays a critical role in embryonic development through affecting various tissues and cell types, including but not limited to vascular system. Although the authors provided data to show that the structure of vessel is relatively normal in *apoc2* mutants, it is insufficient to prove that *apoc2* and FFA-DHA directly affect haematopoiesis. It remains possible that the haematopoietic phenotype is caused by the delay of the whole embryonic development. This possibility is supported by their data showing that the size of *apoc2* mutant embryos is smaller than that of WT embryos (Fig. 1E and F, Fig. 2L and Fig. 3B showed that the body length and head size of *apoc2* mutants are smaller than WT embryos). Indeed, after DHA treatment of *apoc2* mutants, both the haematopoietic and the body size defects were rescued at the same time (Fig. S5 A and S9 B), showing that the general developmental delay and haematopoietic defect cannot be disassociated. To prove the direct function of FFA-DHA in haematopoiesis, the authors should develop a method in which they could supply FFA-DHA specifically into HSCs in *apoc2* mutants to see whether it's sufficient to rescue the haematopoietic defect. A more specific assay will be necessary to clarify this issue.

2. Because lipid metabolism plays a critical role in general development of animals, the current data could not distinguish whether FFA-DHA have a specific role in hematopoiesis. It is thus important to uncover the molecular basis underlying the action of FFA-DHA on haematopoiesis, which, I believe, may provide a definitive evidence demonstrating that FFA-DHA indeed have a specific role in haematopoiesis.

Reviewer #5 (Remarks to the Author):

When dealing with general pathways like lipids, tissue specific effects can be difficult to assess because of embryonic developmental delays. Here the authors have really done a good job of matching the blood vessel timing, so I think the conclusions hold. I think the rescue experiments are quite good here and nailing the DHA as the mechanism.

RESPONSE:

We were very pleased to read Reviewer #1's comment that the authors were extremely responsive to the issues raised in the original review and nicely addressed ALL of the concerns. We are also grateful to Reviewer #5 who served as an adjudicator and confirmed the validity of our conclusions based on the data presented in the manuscript.